# C2IQL: Constraint-Conditioned Implicit Q-learning for Safe Offline Reinforcement Learning

Zifan Liu [1]   Xinran Li [1]   Jun Zhang [1]

## Abstract

Safe offline reinforcement learning aims to develop policies that maximize cumulative rewards while satisfying safety constraints without the need for risky online interaction. However, existing methods often struggle with the out-of-distribution (OOD) problem, leading to potentially unsafe and suboptimal policies. To address this issue, we first propose Constrained Implicit Q-learning (CIQL), a novel algorithm designed to avoid the OOD problem. In particular, CIQL expands the implicit update of reward value functions to constrained settings and then estimates cost value functions under the same implicit policy. Despite its advantages, the further performance improvement of CIQL is still hindered by the inaccurate discounted approximations of constraints. Thus, we further propose Constraint-Conditioned Implicit Q-learning (C2IQL). Building upon CIQL, C2IQL employs a cost reconstruction model to derive non-discounted cumulative costs from discounted values and incorporates a flexible, constraint-conditioned mechanism to accommodate dynamic safety constraints. Experiment results on DSRL benchmarks demonstrate the superiority of C2IQL compared to baseline methods in achieving higher rewards while guaranteeing safety constraints under different threshold conditions.

## 1. Introduction

Reinforcement learning (RL) (Sutton & Barto, 2018) has emerged as a robust framework to solve decision-making problems through constant interactions with environments. However, this "trial and error" approach may not be appli-

cable when such interactions are costly, risky, or impractical, such as in autonomous driving (Fang et al., 2022) and robotics (Weerakoon et al., 2024). To address this challenge, offline RL (Fujimoto et al., 2019) has been proposed, focusing on learning policies from pre-existing datasets without further environmental interaction. A significant challenge that arises in this offline setting is the out-of-distribution (OOD) problem, where an agent may inaccurately estimate the value of state-action pairs not found in the dataset. Various methods have been developed to tackle this challenge, such as regularization (Fujimoto et al., 2019) and constrained bootstrapping (Wu et al., 2020).

While showing impressive achievements in reward-only tasks, offline RL may lose its effectiveness in safety-critical scenarios. For example, unsafe operations could harm patients in healthcare (Den Hengst et al., 2022), unsafe driving styles may lead to accidents (Fang et al., 2022), and unsafe decisions may incur additional costs in financial investments (Froot et al., 1993). In these situations, safety is categorized into the safe exploration process that prevents agents from exploring risky states or actions, and safe optimal criterion that ensures constraint satisfaction of target policies (García & Fernández, 2015). Although offline RL naturally addresses the safe exploration process by avoiding online interaction, it is challenging to satisfy the safe optimal criterion in offline settings due to the OOD problem (Zheng et al., 2024). Thus, safe offline RL (SORL) (Le et al., 2019), which incorporates safety constraints into the learning process, has been proposed by combining safe online RL algorithms and offline RL algorithms. While existing SORL methods (Xu et al., 2022; Lee et al., 2022) have made progress in mitigating the OOD problem and improving the constraint satisfaction ability, further improvement is impeded by two key drawbacks: (1) the inability to avoid the OOD problem in offline RL completely, and (2) the lack of accurate and flexible constraint handling in safe RL. We will elaborate on these in the following:

From the perspective of offline RL, existing SORL methods that leverage OOD detection or regularization can only mitigate the OOD problem but not avoid it. Specially, the agent remains susceptible to encountering unseen state-action pairs during model updates, which can lead to unsafe poli-

[1]Department of Electronic and Computer Engineering, The Hong Kong University of Science and Technology, Hong Kong SAR, China. Correspondence to: Jun Zhang <eejzhang@ust.hk>.

*Proceedings of the 42nd International Conference on Machine Learning*, Vancouver, Canada. PMLR 267, 2025. Copyright 2025 by the author(s).

cies. For instance, some approaches (Fujimoto et al., 2019) train a model for OOD detection but the detection models themselves also introduce errors. Other methods (Kostrikov et al., 2021; Xu et al., 2022), which restrict the learned policy to be close to the behavior policy, can result in constraint violation or suboptimal policy since the behavior policy for collecting data is usually unknown. Implicit Q-learning (IQL) (Kostrikov et al., 2022) offers a potential solution to effectively avoiding the OOD problem. Nonetheless, it is challenging to explicitly achieve both reward maximization and constraint satisfaction since the policy is implicitly hidden in the value function.

From the perspective of safe RL, we identify two key shortcomings that arise from the previous constraint formation: (1) inaccurate discounted formulation of constraints and (2) fixed constraint thresholds that fail to adapt to the dynamic requirements imposed by the environments as well as the long-horizon task nature. These problems can hinder the agent's performance, preventing it from constraint satisfaction or reward maximization. While some methods, such as average-constrained policy optimization (Agnihotri et al., 2024), attempt to address the first problem by solving the constrained Markov decision process on average settings rather than discounted settings, they are mainly suitable for on-policy methods, where the behavior policy and the target policy are the same, making them less applicable to offline RL. Furthermore, most RL-based methods have not adequately addressed the challenge of adapting to dynamic constraint thresholds, which is crucial for long-horizon tasks and varying environmental requirements.

To address the above challenges, this paper proposes Constraint-Conditioned Implicit Q-learning (C2IQL), which is trained within the pre-collected dataset distribution. In particular, C2IQL contains a cost reconstruction model and is equipped with constraint-conditioned ability for better constraint satisfaction and reward maximization. The key contributions are summarized as follows:

(1) From the offline perspective, we first propose Constrained IQL (CIQL) to address the OOD problem in constrained settings. Specifically, we leverage the constraint-penalized update to expand the reward value function in IQL and re-derive cost value functions by generalizing IQL so that both value functions can be updated under the same implicit policy without explicitly extracting it.

(2) From the safety perspective, we improve CIQL and propose C2IQL by addressing the following two key issues regarding the inaccurate constraint thresholds:

- To obtain accurate constraint estimations, we propose to train a cost reconstruction model to reconstruct non-discounted cumulative costs from the corresponding discounted values.

- To handle flexible threshold requirement, we draw inspiration from the goal-conditioned RL method and incorporate constraint-conditioned ability into CIQL. This allows C2IQL to allocate the cost budgets more effectively throughout the task horizon and cope with a wider range of threshold requirements for a better balance between reward maximization and constraint satisfaction.

(3) We evaluate the proposed C2IQL on *Bullet-Safety-Gym* (Gronauer, 2022) and *SafetyGymnasium* (Ji et al., 2023) with DSRL datasets under different threshold conditions. Our empirical results and comprehensive ablation studies demonstrate C2IQL's superiority in both constraint satisfaction and reward maximization.

## 2. Background

In this section, we briefly introduce the constrained optimization problem in safe RL and provide essential background on the constraint-penalized method of CPQ, which will be adopted to update the constrained value function, and the principle of IQL to avoid the OOD problem in offline RL, which we generalize to constrained settings in our method. At last, we provide an overview of related works of safe RL, offline RL, and SORL respectively.

### 2.1. Problem Settings

**Constrained Markov Decision Process (CMDP):** CMDP (Altman, 1998) is a typical formulation of safe RL defined by $\mathcal{M} = (\mathcal{S}, \mathcal{A}, \mathcal{P}, r, c, \gamma)$, where $\mathcal{S}$ is the state space, $\mathcal{A}$ is the action space, $\mathcal{P}(s'|s, a)$ is the transition function, $r(s, a)$ and $c(s, a)$ are the reward and cost functions, and $\gamma$ is the discount factor. In safe RL, we aim to find a policy $\pi(a|s)$ that maximizes the performance measure while satisfying the safety constraints. In episodic tasks, where the trajectory is denoted as $\tau = \{(s_0, a_0, r_0, c_0), ..., (s_T, a_T, r_T, c_T)|r_t = r(s_t, a_t), c_t = c(s_t, a_t), s_{t+1} \sim \mathcal{P}(s_{t+1}|s_t, a_t)\}$, with $T$ being the trajectory length, the optimization problem of safe RL can be written as:

$$
\begin{aligned}
\max_{\pi} \; & \mathbb{E}_{\tau \sim \pi}[R(\tau)], \\
s.t. \; & \mathbb{E}_{\tau \sim \pi}[C(\tau)] \leq L,
\end{aligned}
\tag{1}
$$

where $R(\tau) = \sum_{t=0}^{T} r_t$ is the cumulative reward, $C(\tau) = \sum_{t=0}^{T} c_t$ is the cumulative cost, and $L$ is the constraint threshold.

However, not all environments have explicit termination signals and the estimate of the cumulative reward/cost via the Bellman equation can easily lead to maximizing an infinite value. To ensure convergence and improve stability, discounting is introduced for a general notation for episodic and continuing tasks, which can be proven to have instance dependent sample complexity bound (Jiang & Ye, 2024)

with a discount factor. In this case, the objective can be rewritten as:

$$\max_{\pi} \ V_r^{\pi}(\tau),$$
$$s.t. \ V_c^{\pi}(\tau) \leq \tilde{L}, \quad (2)$$

where $V_r^{\pi}(\tau) = \mathbb{E}_{\tau \sim \pi}[\sum_{t=0}^{\infty} \gamma^t r_t]$ denotes the (reward) value function, $V_c^{\pi}(\tau) = \mathbb{E}_{\tau \sim \pi}[\sum_{t=0}^{\infty} \gamma^t c_t]$ denotes the cost value function and $\tilde{L} = \frac{L}{T} \cdot \frac{1-\gamma^T}{1-\gamma}$ is an average approximation version for $L$. In SORL, the agent is trained with a fixed and pre-collected dataset $D = \{(s, a, r, c, s')\}$ under unknown behavior policies.

## 2.2. Constraints Penalized Q-learning (CPQ)

CPQ is a kind of SORL method updated in a shielding way. The update of CPQ (Xu et al., 2022) consists of three steps. First, CPQ trains a Conditional Variational AutoEncoder (CVAE) (Sohn et al., 2015) to detect OOD state-action pairs and manually train their cost Q-values ($Q_c^{\pi}$) to be larger than the constraint threshold. Second, CPQ proposes constraints penalized update of the reward Q-values ($Q_r^{\pi}$), which is:

$$Q_r^{\pi}(s_t, a_t) = r_t + \gamma \mathbb{E}_{a_{t+1} \sim \pi}[$$
$$\mathbb{1}(Q_c^{\pi}(s_{t+1}, a_{t+1}) \leq \tilde{L}) \, Q_r^{\pi}(s_{t+1}, a_{t+1})], \quad (3)$$

where $\mathbb{1}(\cdot)$ is the indicator function. Finally, the policy is optimized under the constraint, which is:

$$\pi^* = \arg\max_{\pi} \mathbb{E}_{a \sim \pi}[\mathbb{1}(Q_c^{\pi}(s, a) \leq \tilde{L}) \, Q_r^{\pi}(s, a)]. \quad (4)$$

However, CPQ fails to satisfy the constraint as the CVAE model may fail to detect OOD pairs in some cases.

## 2.3. Implicit Q-learning (IQL)

**Implicit Q-learning (IQL) (Kostrikov et al., 2022).** IQL addresses the fundamental OOD challenge in offline Q-learning by restricting the maximization operation within the dataset support. This OOD problem arises during value bootstrapping when standard Q-learning attempts to maximize over actions that may not exist in the offline dataset:

$$Q_r^{\pi}(s_t, a_t) = r_t + \gamma \max_a Q_r^{\pi}(s_{t+1}, a). \quad (5)$$

Directly taking the maximum within the dataset is sensitive to outliers (lucky samples) in the dataset and can lead to overestimation. Thus, IQL introduces a value function based on expectile regression that estimates a conservative expectile of the Q-value distribution:

$$L_V^{\kappa} = \mathbb{E}_{(s,a) \sim D}[L_2^{\kappa}(Q_r^{\pi}(s, a) - V_r^{\pi}(s)], \quad (6)$$

where $L_2^{\kappa}(u) = |\kappa - \mathbb{1}(u < 0)|u^2$ and $\kappa \in (0, 1)$.

Then the reward Q-value function is updated as:

$$Q_r^{\pi}(s_t, a_t) = r_t + \gamma V_r^{\pi}(s_{t+1}). \quad (7)$$

Since the policy is implicitly hidden in the reward value and Q-value functions, IQL utilizes Advantage Weighted Regression (AWR) to extract the policy:

$$L_{\pi} = \mathbb{E}_{(s,a) \sim D}[\exp(\alpha(Q_r^{\pi}(s, a) - V_r^{\pi}(s))) \log \pi(a|s)], \quad (8)$$

where $\alpha \in (0, \infty)$ is a temperature parameter to balance Q-learning optimization and behavior cloning.

**Implicit Diffusion Q-learning (IDQL) (Hansen-Estruch et al., 2023).** IDQL generalizes the expectile regression in IQL to any arbitrary convex function $f$ with $f'(0) = 0$ and re-derives IQL as a standard actor-critic algorithm, in which the optimal value function is given by

$$V_r^*(s) = \arg\min_{V_r^{\pi_{\text{imp}}}} \mathbb{E}_{a \sim \mu(a|s)}[f(Q_r^{\pi_{\text{imp}}}(s, a) - V_r^{\pi_{\text{imp}}}(s))], \quad (9)$$

where $\mu(a|s)$ is the behavior policy and the expectile regression in IQL emerges as a special case when $f(\cdot) = L_2^{\kappa}(\cdot)$.

While policy extraction through AWR in IQL is empirically effective, IDQL shows that this process can be derived from first principles. Based on Equation (9), IDQL proves that the learned implicit policy follows:

$$\pi_{\text{imp}}^* \propto \frac{\mu(a|s)|f'(Q_r^{\pi_{\text{imp}}}(s, a) - V_r^*(s))|}{|Q_r^{\pi_{\text{imp}}}(s, a) - V_r^*(s)|}, \quad (10)$$

where $f' = \frac{\partial f}{\partial V_r^{\pi_{\text{imp}}}}$, and AWR in IQL is another special case of $f$, called exponential regression.

Since the policy is implicitly hidden in the value function, it is difficult to update the cost value function under the same policy for IQL-based methods without extracting it when extending to constrained settings.

## 2.4. Related Work

**Offline RL:** Offline RL aims to optimize a policy on a pre-collected dataset to avoid costly interaction with the environment. A fundamental challenge in offline RL is the OOD problem. Many methods have been proposed to mitigate it by regularizing the optimized policy close to the behavior policy. For example, BCQ (Fujimoto et al., 2019) achieves this by training a generative model as the behavior policy; BEAR (Wu et al., 2020) proposes a constrained bootstrapping operation to reduce the accumulated OOD error; CQL (Kumar et al., 2020) forces the values of OOD state-action pairs to be conservative. Instead of mitigating the OOD problem, IQL (Kostrikov et al., 2022) tries to avoid it by employing expectile regression to learn Q-values within the dataset. More recently, some works formulate offline RL as a return-conditioned generation problem and address it using generative models, such as Decision Transformer (DT) (Chen et al., 2021; Zheng et al., 2022; Wu et al., 2024), and Decision Convformer (DC) (Kim et al., 2024).

**Safe RL:** Safe RL aims to maximize future rewards while satisfying safety constraints (García & Fernández, 2015). Primal-Dual Optimization (PDO) and Constrained Policy Optimization (CPO) are two common approaches in safe RL. PDO (Chow et al., 2018; Ding et al., 2020) leverages Lagrangian multiplier methods to update the primal and dual objectives respectively. Stooke et al. (2020) propose to incorporate the PID control to address the instability issue of PDO. CPO (Achiam et al., 2017; Yang et al., 2020) takes a different route by transforming the constraint into a surrogate objective and updating with it through second-order approximation. To reduce the computation cost of CPO, CUP (Yang et al., 2022) was proposed to optimize the surrogate function with first-order approximation directly. Although PDO and CPO show great promise in online settings, it is challenging for them to handle the OOD problem in offline settings due to their reliance on sampling actions under current policies.

**Safe Offline RL:** Safe offline RL (SORL) integrates safety constraints into offline learning settings, addressing both safety requirements and limited online interaction with the environment. This emerging field has spawned two main approaches: RL-based methods and generative methods. CPQ (Xu et al., 2022) is a representative RL-based method that trains a CVAE model to mitigate the OOD problem based on the constraint-penalized method. COptiDICE (Lee et al., 2022) utilizes stationary distribution correction to mitigate the distributional shift problem. OASIS (Yao et al., 2024) investigates the influence of the unknown behavior policy and provides a dataset distribution shaping method to address this problem. The alternative generative approaches (Liu et al., 2023b) transform SORL into a goal-conditioned generative problem, where expected reward and cost returns are set as the input of the generative models. Nevertheless, these methods struggle to balance the reward and cost conditions, making it hard to maximize the reward while satisfying the constraint. Different from the above two lines of research, our proposed method C2IQL is an RL-based method equipped with the IQL-style update to avoid the OOD problem and can achieve both constraint satisfaction and reward maximization.

## 3. Method

In this section, we introduce a novel algorithm CIQL that re-derives IQL within constrained settings to address the OOD problem and propose an improved version, named C2IQL, by further addressing the inaccurate and inflexible estimation of constraints.

In particular, we will illustrate (1) how to obtain an SORL algorithm based on the constraint-penalized method that avoids the OOD problem by extending IQL to CIQL; (2) Why discounted formulation on the costs is inaccurate and

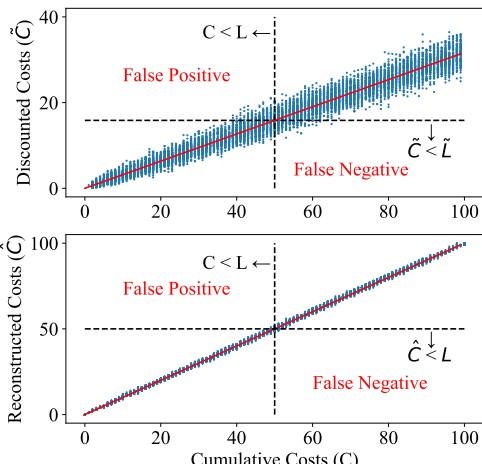

*Figure 1.* Relationship between the discounted costs $\tilde{C}$, reconstructed costs $\hat{C}$ and the cumulative cost $C$ of randomly generated cost trajectories.

what consequences will they lead to; (3) How to address (2) with a cost reconstruction model and incorporate it into CIQL; (4) How to integrate the constraint-conditioned ability into CIQL to improve flexibility; and (5) Practical implementation of our proposed C2IQL algorithm.

### 3.1. Constrained Implicit Q-learning (CIQL)

To derive a concrete CIQL algorithm, we need to answer three questions: **First, how to update the constrained reward value function following the IQL style?** To address this problem, CIQL formulates a constraint-penalized reward Q-value function following CPQ and utilizes a value function with expectile regression to approximate the maximized Q-value function in the Bellman backup procedure. **Second, how to update the cost value function under the same implicit policy since it is hidden in the reward value function?** To address this problem, we rederive CIQL and obtain the formulation of the implicit policy in **Theorem 1** following IDQL, and then derive the formulation of the cost value function with this implicit policy. **Third, How to extract the policy?** We extract the policy in an expectile way following Equation (18).

**Value Function Estimation.** To address the safety requirements in offline RL, we first propose CIQL, which extends IQL to handle safety constraints while maintaining its OOD-avoidance property. We start by considering how to incorporate the safety constraints in the value function. Among existing safe RL approaches, we adopt the shielding mechanism from CPQ, as it naturally enables value updates without explicit policy representation. By injecting the shielding update to IQL, we obtain the constraint-penalized Q-value function as

$$Q_r^\pi(s_t, a_t) = r_t + \gamma \max_a Q_{r|c}^\pi(s_{t+1}, a), \quad (11)$$

where $Q_{r|c}^\pi(s, a) = \mathbb{1}(Q_c^\pi(s, a) \leq \tilde{L}) \cdot Q_r^\pi(s, a)$.

Similar to IQL, we use a value function to approximate an expectile of the constraint-penalized Q-value and the loss function of the value function is given as

$$L_{V_r}^\kappa = \mathbb{E}_{(s,a)\sim D}[L_2^\kappa(Q_{r|c}^\pi(s,a) - V_r^\pi(s)], \qquad (12)$$

where $L_2^\kappa(u) = |\kappa - \mathbb{1}(u < 0)|u^2$ and $\kappa \in (0,1)$.

More generally, the expectile regression loss $L_2^\kappa(u)$ can be replaced by any arbitrary convex function $f(u)$ with $f'(0) = 0$ (Hansen-Estruch et al., 2023). In this case, Equation (12) is generalized to

$$V_r^\pi(s) = \arg\min_{V_r^\pi(s)} \mathbb{E}_{(s,a)\sim D}[f(Q_{r|c}^\pi(s,a) - V_r^\pi(s))]. \qquad (13)$$

By now, we have obtained the formulation and loss function of the constraint-penalized reward value function given the cost Q-value function $Q_c^\pi(s,a)$. The key issue is how to update $Q_c^\pi(s,a)$ in $Q_{r|c}^\pi(s,a)$, where the policy $\pi$ is implicitly hidden in $V_r^\pi(s)$ and $Q_r^\pi(s,a)$. One straightforward solution is to extract the policy and then use it to update the cost Q-value function. However, it will result in the OOD problem since the update based on the extracted policy may fall out of the dataset distribution. To maintain the OOD-avoiding property, we need to derive an update method implicitly following the policy in the reward (Q-) value function. By extending the results in IDQL, we have:

**Theorem 1.** *For every state $s$ and any convex loss function $f$ with $f'(0) = 0$, the solution to Equation (13) is also a solution to the optimization problem in Equation (14) where the implicit policy is a reweight of the behavior policy.*

$$V_r^\pi(s) = \arg\min_{V_r^\pi(s)} \mathbb{E}_{a\sim\pi_{\mathrm{imp}}(a|s)}[(Q_{r|c}^\pi(s,a) - V_r^\pi(s))^2], \qquad (14)$$

*where* $\pi_{\mathrm{imp}}(a|s) \propto \frac{\mu(a|s)|f'(Q_{r|c}^\pi(s,a) - V_r^\pi(s))|}{|Q_{r|c}^\pi(s,a) - V_r^\pi(s)|}$.

The detailed proof is shown in Appendix A.1.

**Theorem 1** provides us with a relationship between the constraint-penalized reward function and the corresponding formulation of the implicit policy. When the implicit policy formulation follows Equation (14), the update of the value function under implicit policy is equivalent to the update formulation of the value function in Equation (13) under behavior policy (the policy for collecting the dataset). Thus, we can utilize the formulation of the implicit policy obtained in **Theorem 1** to derive how to update the cost value function based on its definition in **Theorem 2**.

**Theorem 2.** *Based on **Theorem 1**, the cost (Q-) value functions are specified as Equation (15) and Equation (16).*

$$V_c^\pi(s) = \mathbb{E}_{a\sim\pi_{\mathrm{imp}}(a|s)}[Q_c^\pi(s,a)] \\ = \mathbb{E}_{a\sim\mu(a|s)}[M \cdot Q_c^\pi(s,a)]. \qquad (15)$$

**Algorithm 1** Cost Reconstruction Model

1: **Network:** Initialize the cost reconstruction model $R^c$.
2: **Parameter:** Set discount factors as $\{\gamma_1, \gamma_2, ..., \gamma_m\}$.
3: Calculate the cumulative cost $C^D$ and discounted cumulative cost under different discount factors $C^{D,\gamma_1}, ..., C^{D,\gamma_m}$ for each trajectory in $D$.
4: **for** iteration$= 0, ..., N$ **do**
5:    **Update with data from $D$:**
6:    Sample $\{(C^{D,\gamma_1}, ..., C^{D,\gamma_m}, C^D)\} \sim D$.
7:    $L_{R^c} = MSE(R^c(C^{D,\gamma_1}, ..., C^{D,\gamma_m}), C^D)$.
8:    **Update with randomly generated data:**
9:    Randomly generate $\{(c_0, ..., c_T)\}$.
10:    **for** all trajectories $\in \{(c_0, ..., c_T)\}$ **do**
11:      $C^R = \sum_{t=0}^T c_t$.
12:      $C^{R,\gamma_i} = \sum_{t=0}^T \gamma_i{}^t c_t$ for $i = 1, ..., m$.
13:    **end for**
14:    $L_{R^c} = MSE(R^c(C^{R,\gamma_1}, ..., C^{R,\gamma_m}), C^R)$.
15: **end for**

*where* $M = \frac{|f'(Q_{r|c}^\pi(s,a) - V_r^\pi(s))|}{|Q_{r|c}^\pi(s,a) - V_r^\pi(s)|}$.

$$Q_c^\pi(s,a) = c(s,a) + \gamma V_c^\pi(s). \qquad (16)$$

The detailed proof is shown in Appendix A.2. Accordingly, the loss to update the cost value function is given as:

$$L_{V_c} = \mathbb{E}_{a\sim\pi_{\mathrm{imp}}(a|s)}[(Q_c^\pi(s,a) - V_c^\pi(s))^2] \\ = \mathbb{E}_{a\sim\mu(a|s)}[M \cdot (Q_c^\pi(s,a) - V_c^\pi(s))^2], \qquad (17)$$

In the special case of $f(u) = L_2^\kappa(u)$, we have $M = |\kappa - \mathbb{1}(Q_{r|c}^\pi(s,a) - V_r^\pi(s) < 0)|$.

**Policy Extraction.** We follow **Theorem 1** and extract the policy in an expectile way with the loss function:

$$L_\pi = \mathbb{E}_{a\sim\mu(a|s)}[M \cdot \log\pi(a|s)], \qquad (18)$$

where $M = |\kappa - \mathbb{1}(Q_{r|c}^\pi(s,a) - V_r^\pi(s) < 0)|$.

### 3.2. Motivation for Non-discounted Cost Reconstruction

While introducing discount factors enhances convergence and stability, it causes a mismatch between the original non-discounted constraint in Equation (1) and their discounted approximations in Equation (2). Such mismatch becomes a pronounced problem in safe offline RL because the discounted cumulative cost depends not only on the accumulated cost but also crucially on the temporal distribution of costs within a trajectory.

To demonstrate this problem, we randomly generate various cost trajectories $\{\tau_i = (c_0^i, ..., c_{300}^i)|i = 1, ..., 10000; c_t^i \in \{0,1\}; C(\tau^i) \le 100\}$ and calculate the cumulative cost $C$ and discounted cumulative cost $\tilde{C}$ for each trajectory, as

**Algorithm 2** C2IQL

1: **Network:** Initialize value function $V_r^\pi(s, \hat{L})$, cost value function $V_c^\pi(s, \hat{L})$, Q-value function $Q_r^\pi(s, a, \hat{L})$, cost Q-value function $Q_c^\pi(s, a, \hat{L})$, policy $\pi(a|s, \hat{L})$, cost reconstruction model $R^c$.

2: **Parameter:** Initialize discount factors $\gamma$ for reward and $\hat{\gamma} = \{\gamma_1, \gamma_2, ..., \gamma_m\}$ for cost, expectile parameter $\kappa_1$ and $\kappa_2$, threshold set $\mathcal{L}$.

3: **for** iteration$= 0, ..., N$ **do**

4:     Sample transitions $\{(s, a, r, c, s')\} \sim D$.

5:     Randomly sample a threshold $\hat{L} \in \mathcal{L}$.

6:     **Cost Reconstruction:** Obtain discounted cost values and reconstruct the non-discounted value in Section 3.3.

7:     $Q_c^\pi(s, a, \hat{L}) = Q_c^{\pi, \gamma_1}(s, a, \hat{L}), ..., Q_c^{\pi, \gamma_m}(s, a, \hat{L})$.

8:     $\hat{Q}_c^\pi(s, a, \hat{L}) = R^c(Q_c^\pi(s, a, \hat{L}))$.

9:     **Costraint Penalization:** Obtain the constraint-penalized Q-value function with Equation (11).

10:     $Q_{r|c}^\pi(s, a, \hat{L}) = \mathbb{1}(\hat{Q}_c^\pi(s, a, \hat{L}) \le \hat{L})Q_r^\pi(s, a, \hat{L})$.

11:     **Update value function:** Update value function with Equation (12).

12:     $L_{V_r}^{\kappa_1} = \mathbb{E}_{(s,a)\sim D}[L_2^{\kappa_1}(Q_{r|c}^\pi(s, a, \hat{L}) - V_r^\pi(s, \hat{L})]$.

13:     **Update cost value function:** Update cost value function with Equation (17).

14:     $L_{V_c} = \mathbb{E}_{(s,a)\sim D}[|\kappa_1 - \mathbb{1}(Q_{r|c}^\pi(s, a, \hat{L}) - V_r^\pi(s, \hat{L}) < 0)| \cdot (Q_c^\pi(s, a, \hat{L}) - V_c^\pi(s, \hat{L}))^2]$.

15:     **Update Q-value function.**

16:     $L_{Q_r} = \mathbb{E}_{(s,a)\sim D}[(r + \gamma V_r^\pi(s', \hat{L}) - Q_r^\pi(s, a, \hat{L}))^2]$.

17:     **Update cost Q-value function for each $\gamma_i \in \hat{\gamma}$.**

18:     $L_{Q_c} = \mathbb{E}_{(s,a)\sim D}[(c + \hat{\gamma} V_c^\pi(s', \hat{L}) - Q_c^\pi(s, a, \hat{L}))^2]$.

19:     **Policy Extraction.**

20:     $L_\pi = -\mathbb{E}_{(s,a)\sim D}[|\kappa_2 - \mathbb{1}(Q_{r|c}^\pi(s, a, \hat{L}) - V_r^\pi(s, \hat{L}) < 0)| \cdot \log \pi(a|s, \hat{L})]$.

21: **end for**

---

shown in the top figure of Figure 1. The results reveal that the mapping from cumulative costs to discounted values is many-to-one, creating two types of problematic cases when approximating a constraint $C(\tau^i) \le L$ with $\tilde{C}(\tau^i) \le \tilde{L}$. (1) False Positives: Trajectories with early-concentrated costs may satisfy the original constraint $C(\tau^i) \le L$ but violate the discounted approximation $\tilde{C}(\tau^i) \le \tilde{L}$, leading to overly conservative policies that reject safe trajectories. (2) False Negatives: Trajectories with delayed costs may satisfy the discounted constraint while violating the original one, potentially leading to unsafe policies.

### 3.3. Cost Reconstruction Model

Following the definition of Equation (2), the cost value function ($Q_c^{\pi, \gamma}(s, a)$) is a linear summation of cost functions $c(s, a)$ weighted by $\gamma^t < 1$. Similarly, the non-discounted

cost value ($\hat{Q}_c^\pi(s, a)$) is a linear summation of cost functions $c(s, a)$. To address the discounting-induced mismatch described in Section 3.2, the objective is solving a linear equation system:

$$\text{obtain}_{\{c(.,.)\}} \quad \hat{Q}_c^\pi(s_t, a_t) = \sum_{j=0}^{T} c(s_{t+j}, a_{t+j} \sim \pi),$$

$$s.t. \quad Q_c^{\pi, \gamma}(s_t, a_t) = \sum_{j=0}^{T} \gamma^j c(s_{t+j}, a_{t+j} \sim \pi), \tag{19}$$

where $Q_c^{\pi, \gamma}(s_t, a_t)$ are known values estimated by discounted Q-value functions.

However, solving Equation (19) exactly is challenging due to the massive number of variables. To address this problem, we relax the objective function and constrict the constraint to:

$$\min_{\hat{Q}_c^\pi(s_t, a_t)} [\hat{Q}_c^\pi(s_t, a_t) - \sum_{j=0}^{T} c(s_{t+j}, a_{t+j} \sim \pi)]^2$$

$$s.t. \, Q_c^{\pi, \gamma_i}(s_t, a_t) = \sum_{j=0}^{T} \gamma_i^j c(s_{t+j}, a_{t+j} \sim \pi), \forall i = 1, ..., m \tag{20}$$

Then the goal becomes training an accurate cost reconstruction model, denoted as $R^c$, to estimate the reconstructed cost $\hat{Q}_c^\pi$ from discounted cost values with different discount factors $\{\gamma_1, \gamma_2, ..., \gamma_m\}$:

$$\hat{Q}_c^\pi(s, a) = R^c(Q_c^{\pi, \gamma_1}(s, a), ..., Q_c^{\pi, \gamma_m}(s, a)). \tag{21}$$

Algorithm 1 demonstrates the training process of the reconstruction model. Specifically, we use the MSE loss to update the cost reconstruction model $R^c$ and augment the original dataset with some randomly generated data to improve the generalizability of the model.

The bottom figure of Figure 1 demonstrates the reconstructed value of the generated trajectories in Section 3.2. Compared with the discounted value shown at the top, the false positive and negative trajectories with reconstructed cumulative cost are significantly reduced. Constraint satisfaction with the reconstructed cumulative cost provides a more accurate estimation of the cumulative cost compared to using the discounted value.

### 3.4. Constraint-Conditioned Ability

CIQL employs a fixed threshold criterion $\mathbb{1}(\hat{Q}_c^\pi(s_{t+1}, a) \le L)$ that fails to account for the dynamic nature of accumulated costs. This limitation prevents the agent from adapting its behavior based on the historical cost accumulation and remaining cost budget. For instance, when past costs are low, the agent should be able to pursue more aggressive, reward-maximizing policies within the remaining cost budget. Conversely, when significant costs have already been incurred, the agent should adopt more conservative policies to ensure overall constraint satisfaction. To address

*Table 1.* Normalized evaluation results. The normalized cost threshold is set to 1. Values in (.) represent the standard deviation. Each value represents the average performance over 10 evaluation episodes with 5 seeds and 3 thresholds. **Black** indicates safe results; gray indicates unsafe results; and blue indicates safe and best-performing results. ↑ (↓) incidates that higher (lower) values are better.

| Algorithm | Metric | Tasks | | | | | | | | |
| --- | --- | --- | --- | --- | --- | --- | --- | --- | --- | --- |
| | | Run | | | | Circle | | | | |
| | | Ant | Ball | Car | Drone | Ant | Ball | Car | Drone | Avg |
| BCQ-Lag | reward ↑ | $0.56_{(0.06)}$ | $0.27_{(0.07)}$ | $0.92_{(0.01)}$ | $0.76_{(0.06)}$ | $0.76_{(0.05)}$ | $0.64_{(0.04)}$ | $0.67_{(0.04)}$ | $0.96_{(0.01)}$ | $0.69$ |
| | cost ↓ | $0.28_{(0.12)}$ | $0.39_{(0.25)}$ | $0.10_{(0.30)}$ | $1.62_{(0.23)}$ | $1.73_{(0.21)}$ | $0.98_{(0.09)}$ | $1.23_{(0.23)}$ | $2.09_{(0.04)}$ | $1.05$ |
| BEAR-Lag | reward ↑ | $0.14_{(0.02)}$ | $0.22_{(0.5)}$ | $0.80_{(0.16)}$ | $0.02_{(0.15)}$ | $0.41_{(0.19)}$ | $0.75_{(0.03)}$ | $0.73_{(0.03)}$ | $0.85_{(0.03)}$ | $0.49$ |
| | cost ↓ | $0.02_{(0.03)}$ | $1.97_{(0.31)}$ | $3.71_{(2.53)}$ | $0.89_{(0.59)}$ | $1.10_{(0.43)}$ | $1.18_{(0.10)}$ | $1.50_{(0.15)}$ | $1.59_{(0.20)}$ | $1.49$ |
| COptiDICE | reward ↑ | $0.57_{(0.01)}$ | $0.47_{(0.09)}$ | $0.90_{(0.02)}$ | $0.63_{(0.01)}$ | $0.19_{(0.05)}$ | $0.62_{(0.03)}$ | $0.43_{(0.02)}$ | $0.40_{(0.01)}$ | $0.53$ |
| | cost ↓ | $0.22_{(0.11)}$ | $1.25_{(0.23)}$ | $0.00_{(0.00)}$ | $1.40_{(0.03)}$ | $0.91_{(0.29)}$ | $0.99_{(0.11)}$ | $1.20_{(0.19)}$ | $0.42_{(0.10)}$ | $0.80$ |
| CPQ | reward ↑ | $0.05_{(0.00)}$ | $0.63_{(0.11)}$ | $0.94_{(0.03)}$ | $0.33_{(0.15)}$ | $0.02_{(0.02)}$ | $0.63_{(0.05)}$ | $0.59_{(0.13)}$ | $0.13_{(0.01)}$ | $0.41$ |
| | cost ↓ | $0.00_{(0.00)}$ | $1.41_{(0.41)}$ | $1.42_{(1.14)}$ | $1.62_{(0.41)}$ | $0.06_{(0.09)}$ | $0.87_{(0.13)}$ | $1.03_{(0.38)}$ | $0.24_{(0.19)}$ | $0.83$ |
| FISOR | reward ↑ | $0.18_{(0.03)}$ | $0.24_{(0.01)}$ | $0.80_{(0.01)}$ | $0.20_{(0.05)}$ | $0.11_{(0.03)}$ | $0.25_{(0.02)}$ | $0.23_{(0.05)}$ | $0.46_{(0.03)}$ | $0.31$ |
| | cost ↓ | $0.00_{(0.00)}$ | $0.00_{(0.00)}$ | $0.01_{(0.03)}$ | $0.41_{(0.17)}$ | $0.00_{(0.00)}$ | $0.03_{(0.03)}$ | $0.00_{(0.01)}$ | $0.00_{(0.00)}$ | $0.06$ |
| VOCE | reward ↑ | $0.32_{(0.03)}$ | $0.79_{(0.01)}$ | $0.43_{(0.54)}$ | $0.48_{(0.19)}$ | $0.00_{(0.00)}$ | $0.85_{(0.01)}$ | $0.39_{(0.22)}$ | $0.12_{(0.01)}$ | $0.42$ |
| | cost ↓ | $0.86_{(0.24)}$ | $1.04_{(0.00)}$ | $6.58_{(0.01)}$ | $1.58_{(0.39)}$ | $1.01_{(0.46)}$ | $1.34_{(0.04)}$ | $1.43_{(0.87)}$ | $0.57_{(0.41)}$ | $1.80$ |
| WSAC | reward ↑ | $0.25_{(0.04)}$ | $0.80_{(0.30)}$ | $0.86_{(0.09)}$ | $0.66_{(0.12)}$ | $0.40_{(0.08)}$ | $0.69_{(0.08)}$ | $0.61_{(0.14)}$ | $0.02_{(0.01)}$ | $0.54$ |
| | cost ↓ | $0.18_{(0.08)}$ | $1.98_{(0.33)}$ | $0.40_{(0.08)}$ | $2.52_{(0.20)}$ | $0.98_{(0.12)}$ | $0.78_{(0.15)}$ | $0.51_{(0.14)}$ | $0.45_{(0.28)}$ | $0.97$ |
| CDT | reward ↑ | $0.72_{(0.03)}$ | $0.56_{(0.01)}$ | $0.95_{(0.00)}$ | $0.66_{(0.02)}$ | $0.54_{(0.06)}$ | $0.78_{(0.01)}$ | $0.72_{(0.02)}$ | $0.75_{(0.01)}$ | $0.71$ |
| | cost ↓ | $1.00_{(0.08)}$ | $0.97_{(0.02)}$ | $0.80_{(0.21)}$ | $0.73_{(0.13)}$ | $0.96_{(0.13)}$ | $0.96_{(0.03)}$ | $0.95_{(0.10)}$ | $0.97_{(0.04)}$ | $0.91$ |
| C2IQL | reward ↑ | $0.74_{(0.01)}$ | $0.59_{(0.05)}$ | $0.95_{(0.01)}$ | $0.71_{(0.02)}$ | $0.66_{(0.06)}$ | $0.72_{(0.03)}$ | $0.74_{(0.01)}$ | $0.78_{(0.02)}$ | $0.74$ |
| | cost ↓ | $0.94_{(0.05)}$ | $0.95_{(0.07)}$ | $0.08_{(0.17)}$ | $0.73_{(0.16)}$ | $0.76_{(0.13)}$ | $0.85_{(0.07)}$ | $0.93_{(0.07)}$ | $0.85_{(0.11)}$ | $0.76$ |

this problem, we include possible threshold conditions $\hat{L}$ in the input of policy and value functions in CIQL, such as $V_r^\pi(s, \hat{L})$ and $V_c^\pi(s, \hat{L})$. By incorporating the threshold as an input, the agent can dynamically adjust its policy based on the remaining budget, enabling it to adapt to the evolving cost requirement.

### 3.5. Practical Implementation

To sum up, Algorithm 2 demonstrates the overall training procedure of C2IQL. Notably, we use reconstructed costs in line 10 for Q-values with more accurate constraint penalty. In practice, we update the value function and cost value function with the same expectile parameter $\kappa_1$ and update the policy function with another expectile parameter $\kappa_2$. This design maintains a balance between IQL and IDQL approaches, with $\kappa_2$ serving as a temperature parameter to balance between behavior cloning and maximization of the value function. To reduce the computational complexity, instead of updating all constraint conditions, we predefine a threshold set $\mathcal{L}$ and sample thresholds randomly from it during training. More details of the implementation and hyperparameters used in Section 4 are available in Appendix B.2.

## 4. Experiments

**Environments and Datasets.** We evaluate C2IQL in *Bullet-Safety-Gym* (Gronauer, 2022), which includes various continuous robot locomotion control tasks commonly used in previous works. We select four types of robots: *Ant*, *Ball*,

*Car*, and *Drone*; and two kinds of tasks: *Run* and *Circle*. We use the DSRL (Liu et al., 2023a) dataset, which follows the D4RL (Fu et al., 2020) benchmark format. The details of the environments and more experiments on *SafetyGymnasium* are described in Appendices B.1 and B.3.

**Baselines.** We compare proposed C2IQL with the following baseline methods: (1) Primal-dual optimization (Stooke et al., 2020): BCQ-Lag and BEAR-Lag; (2) Constraint penalized method: CPQ (Xu et al., 2022); (3) Distribution correction estimation: COptiDICE (Lee et al., 2022); (4) Sequential modeling algorithms: CDT (Liu et al., 2023b) and FISOR (Zheng et al., 2024); (5) Variational optimization with conservative estimation: VOCE (Guan et al., 2023); (6) Weighted safe actor-critic: WSAC (Wei et al., 2024).

**Metrics.** We evaluate performance using normalized reward and cost returns. The normalized reward return is defined by $R_{\text{norm}} = \frac{R_\pi - R_{\min}}{R_{\max} - R_{\min}}$, where $R_\pi$ is the reward return under policy $\pi$, $R_{\min}$ and $R_{\max}$ are the minimum and maximum reward returns in dataset. The normalized cost return is defined by $C_{\text{norm}} = \frac{C_\pi}{L}$, where $C_\pi$ is the cost return under policy $\pi$ and $L$ is the selected threshold. To provide a comprehensive evaluation, we assess the performance of the algorithms across three cost thresholds: "small", "middle", and "large", calibrated to each environment's cost range. This evaluation scheme allows us to analyze both constraint satisfaction and reward maximization capabilities of algorithms across varying degrees of constraints. Detailed values are provided in Appendix B.2.

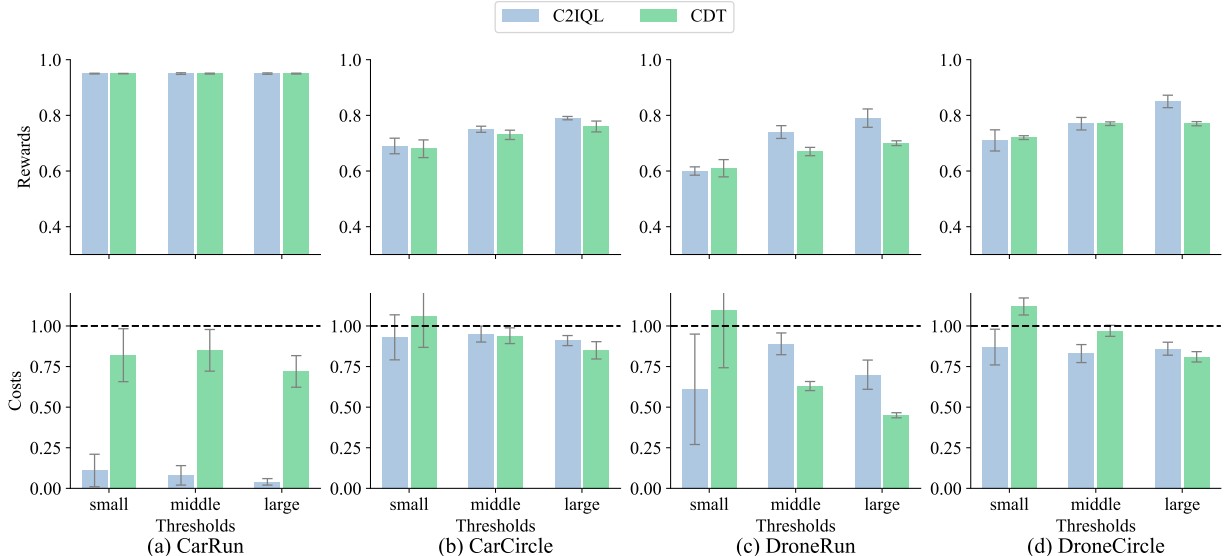

*Figure 2.* Normalized evaluation results comparison between C2IQL and CDT with varying constraint thresholds.

## 4.1. Main Results

Table 1 presents the evaluation results. C2IQL achieves the best and safe performance in most tasks, demonstrating its superiority in reward maximization and constraint satisfaction. Most baselines except FISOR and CDT generally suffer from constraint violations. Even when they show safe behaviors, the performance is inferior compared to C2IQL. FISOR, on the other hand, achieves safe results across all environments but exhibits overly conservative behavior due to its hard constraints property, which limits its ability to achieve high rewards.

Although CDT achieves similar results to C2IQL in Table 1, a closer examination of their performance under different safety thresholds in Figure 2 reveals that C2IQL can flexibly adjust its policy when dealing with different safety thresholds, while CDT does not. CDT fails to ensure safety in certain environments for small thresholds, whereas C2IQL consistently achieves safe results. In contrast, CDT cannot maximize rewards as effectively as C2IQL for large thresholds, even though both methods ensure safety. Notably, as the threshold increases, C2IQL progressively improves its performance, while CDT encounters limitations in specific environments such as DroneCircle.

## 4.2. Ablation Study

To evaluate how much the cost reconstruction model and constraint-conditioned method contribute to the performance gain, we conduct an ablation study with the following variants of C2IQL:

- **C2IQL w/o CR:** C2IQL without Cost Reconstruction module introduced in Section 3.3. The discounted threshold follows the average approximation version, which is $\tilde{L} = \frac{L}{T} \cdot \frac{1-\gamma^T}{1-\gamma}$.

- **C2IQL w/o CC:** C2IQL without Constraint-Conditioned formulation introduced in Section 3.4.

- **CIQL**: The algorithm proposed in Section 3.1, where both the cost reconstruction model and constraint-conditioned ability are not included.

As shown in Figure 3, C2IQL w/o CR demonstrates inferior reward performance compared with C2IQL even though C2IQL w/o CR reaches the cumulative costs close to the threshold due to the constraint-conditioned ability. This indicates that the cost reconstruction model helps to improve reward maximization. On the other hand, C2IQL w/o CC exhibits conservative behavior in terms of costs compared with C2IQL, which indicates that the constraint-conditioned formulation helps to improve the constraint satisfaction ability. CIQL, which lacks both the cost reconstruction model and constraint-conditioned ability, converges to a safe but over-conservative policy for all thresholds. While the safety of CIQL benefits from avoiding the OOD problem and the over-conservative policy is caused by the inaccurate discounted constraint and the absence of constraint-conditioned ability. As a result, CIQL can only achieve very low performance with nearly zero cost, regardless of the scenario. Among all the variants, only C2IQL is able to achieve the highest rewards with the closest costs to the thresholds, which demonstrates the combination of the cost reconstruction model and constraint-conditioned ability enables C2IQL to strike a balance between maximizing rewards and satisfying constraints effectively.

## 4.3. Hyper-parameter Analysis

We conduct analysis of two hyperparameters: the expectile parameter on the value function and the cost value function $\kappa_1$ and the expectile parameter on policy extraction $\kappa_2$, shown in Figure 4. $\kappa_1$ primarily affects constraint satisfac-

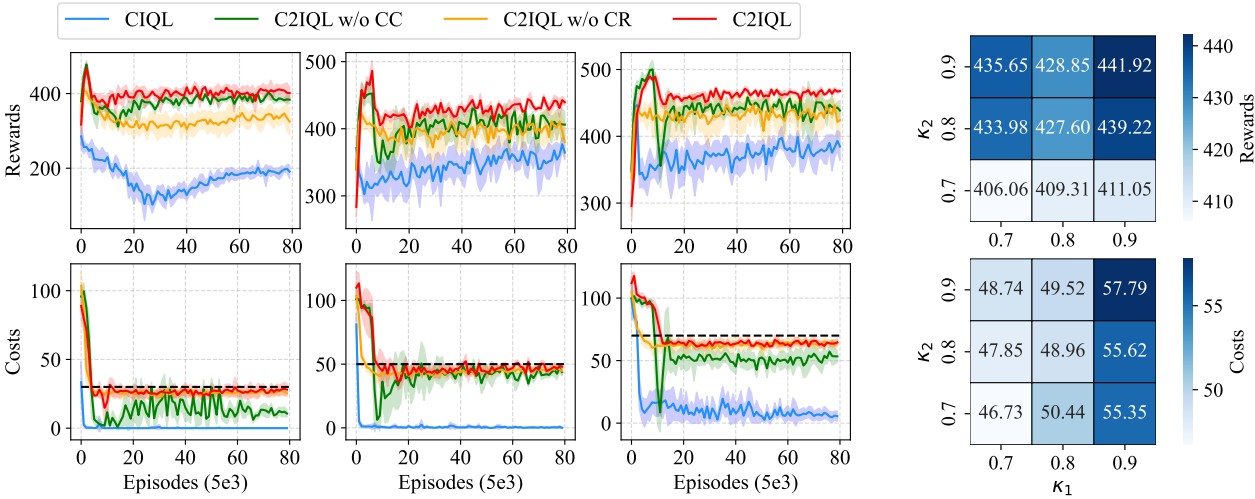

*Figure 3.* Ablation study on *CarCircle* with varying constraint thresholds. The three plots from left to right represent different threshold levels: small = 30, middle = 50, and high = 70. The dashed lines in each plot indicate the corresponding threshold value for that case.

*Figure 4.* Hyperparameter analysis on *CarCircle* with threshold = 50.

*Table 2.* Evaluation results of C2IQL, whose CRM is trained on different noise levels, on task *CarCircle*. The constraint threshold is selected as 50. MSE indicates the training loss of CRM.

| Noise | 0.0 | 0.1 | 0.2 | 0.3 | 0.5 | 1.0 | 2.0 | 4.0 |
|---|---|---|---|---|---|---|---|---|
| MSE | 0.025 | 1.10 | 1.77 | 2.25 | 3.35 | 4.41 | 5.76 | 7.84 |
| reward ↑ | 434.01 | 435.07 | 442.08 | 435.67 | 435.60 | 421.82 | 408.77 | 401.89 |
| cost ↓ | 46.60 | 47.20 | 48.30 | 47.30 | 47.89 | 42.10 | 34.90 | 36.20 |

tion where a smaller $\kappa_1$ leads to smaller costs. Excessive $\kappa_1$ values, such as $0.9$, can cause the value function to be affected by some "luckily safe" transitions, also known as "lucky samples" in IQL, and thus result in an unsafe policy. $\kappa_2$ primarily affects reward maximization where a larger $\kappa_2$ leads to higher performance with little effect on safety. The main reason is that smaller $\kappa_2$ can make the policy closer to behavior cloning, degrading the performance. These findings suggest that selecting a smaller $\kappa_1$ value and a larger $\kappa_2$ value are often a good choice for achieving both safety and high performance.

### 4.4. Robustness of the Cost Reconstruction Model

To validate how the accuracy of CRM influences the performance of C2IQL, we introduce the noise sampled from a normal distribution, $N(0, \text{noise})$, to the input of CRM and train C2IQL on task *CarCircle*. As shown in Table 2, the performance slightly improves with constraint satisfaction as the noise level increases when the added noise is small. This occurs because small amounts of noise act similarly to data augmentation, which enhances the generalization capability of the cost reconstruction model. This is analogous to adding noise to input data in image processing to improve robustness. As noise increases from 1.0 to 4.0, the performance becomes progressively more conservative.

Despite this, the reward and cost metrics degrade gracefully, indicating the robustness of C2IQL.

## 5. Conclusion

In this paper, we introduce C2IQL, a novel approach to SORL that effectively addresses the OOD challenge in constrained settings by leveraging constraint-penalized implicit updates. Our findings also highlight the critical issues associated with inaccurate discounted approximations of constraints and the rigidity of fixed constraint thresholds. By implementing a cost reconstruction model, C2IQL adeptly reconstructs non-discounted cumulative costs from discounted values across varying discount factors. Additionally, the constraint-conditioned mechanism allows for dynamic adjustment of constraint thresholds, significantly enhancing the algorithm's adaptability. These innovations make C2IQL a promising approach for SORL, offering both improved OOD handling, and accurate and dynamic constraint satisfaction abilities, paving the way for more robust and reliable SORL algorithms in the future.

## Acknowledgements

This work was supported by the Hong Kong Research Grants Council under the Areas of Excellence scheme grant

AoE/E-601/22-R and NSFC/RGC Collaborative Research Scheme grant CRS_HKUST603/22.

## Impact Statement

This paper presents work whose goal is to advance the field of Safe Reinforcement Learning in offline settings. There are many potential societal consequences of our work, none of which we feel must be specifically highlighted here.

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

# A. Proof

## A.1. Proof of Theorem 1.

$$V_r^*(s) = \arg\min_{V_r^\pi(s)} \mathbb{E}_{(s,a)\sim D}[f(Q_{r|c}^\pi(s,a) - V_r^\pi(s))]$$

$$= \arg\min_{V_r^\pi(s)} \mathbb{E}_{a\sim\mu(a|s)}[f(Q_{r|c}^\pi(s,a) - V_r^\pi(s))]$$

For convex function $f$, the optimality is achieved when:

$$0 = \frac{\partial}{\partial V_r^\pi(s)}\mathbb{E}_{a\sim\mu(a|s)}[f(Q_{r|c}^\pi(s,a) - V_r^\pi(s))]|_{V_r^\pi(s)=V_r^*(s)}$$

$$= -\mathbb{E}_{a\sim\mu(a|s)}[f'(Q_{r|c}^\pi(s,a) - V_r^*(s))]$$

Due to convexity of $f$ and the assumption $f'(0) = 0$, we have $f'(x) = |f'(x)|\frac{x}{|x|}$.

$$= -\mathbb{E}_{a\sim\mu(a|s)}[|f'(Q_{r|c}^\pi(s,a) - V_r^*(s))| \cdot \frac{Q_{r|c}^\pi(s,a) - V_r^*(s)}{|Q_{r|c}^\pi(s,a) - V_r^*(s)|}]$$

$$= -\int_a \mu(a|s)|f'(Q_{r|c}^\pi(s,a) - V_r^*(s))| \cdot \frac{Q_{r|c}^\pi(s,a) - V_r^*(s)}{|Q_{r|c}^\pi(s,a) - V_r^*(s)|}$$

We then define the implicit policy to be $\pi^{\text{imp}}(a|s) = \mu(a|s)\frac{|f'(Q_{r|c}^\pi(s,a)-V_r^*(s))|}{Z_{\text{imp}}|Q_{r|c}^\pi(s,a)-V_r^*(s)|}$, where $Z_{\text{imp}}$ is a normalization factor to keep the sum of the probability as 1.

$$= -Z_{\text{imp}} \cdot \int_a \pi^{\text{imp}}(a|s)(Q_{r|c}^\pi(s,a) - V_r^*(s))$$

$$= -Z_{\text{imp}} \cdot \mathbb{E}_{\pi^{\text{imp}}(a|s)}[Q_{r|c}^\pi(s,a) - V_r^*(s)]$$

$$= \frac{Z_{\text{imp}}}{2} \cdot \frac{\partial}{\partial V_r^\pi(s)}\mathbb{E}_{\pi^{\text{imp}}(a|s)}[(Q_{r|c}^\pi(s,a) - V_r^*(s))^2]$$

$$= 0$$

This means that $V_r^*(s)$ is also a solution for the optimization problem:

$$\arg\min_{V_r^\pi(s)} \mathbb{E}_{a\sim\pi_{\text{imp}}(a|s)}[(Q_{r|c}^\pi(s,a) - V_r^\pi(s))^2]$$

## A.2. Proof of Theorem 2.

$$V_c^\pi(s) = \mathbb{E}_{a\sim\pi_{\text{imp}}(a|s)}[Q_c^\pi(s,a)]$$

$$0 = \mathbb{E}_{a\sim\pi_{\text{imp}}(a|s)}[Q_c^\pi(s,a) - V_c^\pi(s)]|_{V_c^\pi(s)=V_c^*(s)}$$

$$= \mathbb{E}_{a\sim\pi_{\text{imp}}(a|s)}[Q_c^\pi(s,a) - V_c^*(s)]$$

$$= \int_a \pi^{\text{imp}}(a|s)(Q_c^\pi(s,a) - V_c^*(s))$$

$$= -\int_a \pi^{\text{imp}}(a|s)(Q_c^\pi(s,a) - V_c^*(s))$$

Following the definition of the implicit policy in Theorem 1: $\pi^{\text{imp}}(a|s) = \mu(a|s)\frac{|f'(Q_{r|c}^\pi(s,a)-V_r^*(s))|}{Z_{\text{imp}}|Q_{r|c}^\pi(s,a)-V_r^*(s)|}$, where $Z_{\text{imp}}$ is a normalization factor to keep the sum of the probability as 1.

$$= -\frac{1}{Z_{\text{imp}}}\int_a \mu(a|s)\frac{|f'(Q_{r|c}^\pi(s,a) - V_r^*(s))|}{|Q_{r|c}^\pi(s,a) - V_r^*(s)|}(Q_c^\pi(s,a) - V_c^*(s))$$

$$= -\frac{1}{Z_{\text{imp}}}\mathbb{E}_{a\sim\mu(a|s)}[\frac{|f'(Q_{r|c}^\pi(s,a) - V_r^*(s))|}{|Q_{r|c}^\pi(s,a) - V_r^*(s)|}(Q_c^\pi(s,a) - V_c^*(s))]$$

$$= \frac{1}{2Z_{\text{imp}}}\frac{\partial}{\partial V_c^\pi(s)}\mathbb{E}_{a\sim\mu(a|s)}[\frac{|f'(Q_{r|c}^\pi(s,a) - V_r^*(s))|}{|Q_{r|c}^\pi(s,a) - V_r^*(s)|}(Q_c^\pi(s,a) - V_c^\pi(s))^2]$$

This means that $V_c^*(s)$ is a solution for the optimization problem

$$\arg \min_{V_c^\pi(s)} \mathbb{E}_{a \sim \mu(a|s)} \left[ \frac{|f'(Q_{r|c}^\pi(s,a) - V_r^*(s))|}{|Q_{r|c}^\pi(s,a) - V_r^*(s)|} (Q_c^\pi(s,a) - V_c^\pi(s))^2 \right]$$

If we take $f$ as $L_2^\kappa$, the loss function should be

$$L_{V_c} = \mathbb{E}_{(s,a) \sim D} [|\kappa - \mathbb{1}(Q_{r|c}^\pi(s,a) - V_r^\pi(s) < 0)| \cdot (Q_c^\pi(s,a) - V_c^\pi(s))^2]$$

## B. Experiment Details

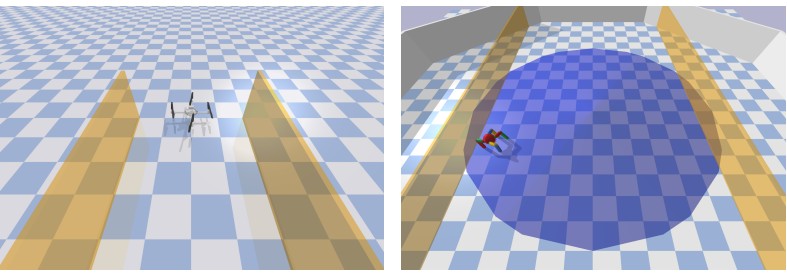

*Figure 5.* Run task.          *Figure 6.* Circle task.

### B.1. Environment Details

*Bullet-Safety-Gym* (Gronauer, 2022) consists of two types of tasks: *Run* and *Circle*. In *Run* tasks shown in Figure 5, the rewards of the agent come from running between two safety boundaries with high speed and the costs come from either reaching the boundaries or exceeding specific velocity related to different robots. In *Circle* tasks shown in Figure 6, the rewards of the agent come from moving circularly within the circle area and the costs come from when the robot leaves the safety zone of the yellow boundaries. Besides two tasks, there are also four kinds of robots: *Ant*, *Ball*, *Car*, and *Drone* with different parameters, shown in Table 3.

*Table 3.* Environment parameters.

| Parameter | Tasks | | | | | | | |
|---|---|---|---|---|---|---|---|---|
| | Run | | | | Circle | | | |
| | Ant | Ball | Car | Drone | Ant | Ball | Car | Drone |
| Episode length (T) | 200 | 100 | 200 | 200 | 500 | 200 | 300 | 300 |
| Action space | 8 | 2 | 2 | 4 | 8 | 2 | 2 | 4 |
| State space | 33 | 7 | 7 | 17 | 34 | 8 | 8 | 18 |
| Cost range | 150 | 80 | 40 | 140 | 200 | 80 | 100 | 100 |

### B.2. Implementation Details on C2IQL

For C2IQL, the structure and most hyperparameters follow IQL (Kostrikov et al., 2022). Table 4 shows the hyperparameters of our algorithms. The discount factor of the reward is fixed at $0.99$ and the number of discount factors for the cost is 3. Table 5 demonstrates the discount factors for cost reconstruction and different thresholds for comprehensive comparison. For most environments, the selection of discount factors are based on the episode length, where $\gamma_1^T \approx 0.05$, $\gamma_2^T \approx 0.1$, and $\gamma_3^T \approx 0.2$, except for *BallRun* and *AntCircle*. We select smaller discount factors for better stability and convergence for these two environments. Besides, we select three thresholds considering the cost range of each environment. In particular, we select a "small" value, which is less than $50\%$; a "middle" value, which is approximately $50\%$; and a "high" value, which is more than $50\%$.

For the cost reconstruction model, we use a 5-layer MLP with hidden dimensions of 512 for each layer. We pre-train the cost reconstruction model for each environment with the offline dataset and randomly generated data. Besides, to maintain

computational efficiency while computing multiple discounted estimates, we employ parameter sharing across different discount factors by extending the cost value and Q-value functions to produce multiple outputs:

$$Q_c^\pi(s, a) = Q_c^{\pi, \gamma_1}(s, a), ..., Q_c^{\pi, \gamma_m}(s, a). \tag{22}$$

To prevent overfitting and improve robustness, we add the noise sampled from a normal distribution $N(0, 0.1)$ to the input. We pre-train the reconstruction model for $1e6$ epochs for each environment.

For the threshold set $\mathcal{L}$ of each environment, we set $\mathcal{L} = \{\text{small}, \text{middle}, \text{high}, \infty\}$ as the constraint condition of C2IQL for the training procedure. In the testing procedure, we scale the cost budget linearly based on horizon $T$, as shown in Equation (23), and then choose the closest value in $\mathcal{L}$ as the constraint condition.

$$\hat{L}_t = C_t^b * T/(T - t), \tag{23}$$

where $C_t^b = L - \sum_{i=0}^{t-1} c_i$ is the cost budget at time step $t$.

*Table 4.* Hyperparameters

| Hyperparameters | Value |
|---|---|
| $\kappa_1$ | 0.7 |
| $\kappa_2$ | 0.9 |
| $\gamma$ (reward) | 0.99 |
| $m$ | 3 |
| Batch size | 512 |
| Learning rate of $V$ | 1e-3 |
| Learning rate of $Q$ | 1e-3 |
| Learning rate of $\pi$ | 3e-4 |
| Training steps | 4e5 |
| Testing frequency | 5e3 |

*Table 5.* Discount factors for cost reconstruction and constraint thresholds for experiments.

| Hyperparameters | Tasks | | | | | | | |
|---|---|---|---|---|---|---|---|---|
| | Run | | | | Circle | | | |
| | Ant | Ball | Car | Drone | Ant | Ball | Car | Drone |
| **Episode length (T)** | **200** | **100** | **200** | **200** | **500** | **200** | **300** | **300** |
| $\gamma_1$ | 0.985 | 0.95 | 0.985 | 0.985 | 0.99 | 0.985 | 0.99 | 0.99 |
| $\gamma_2$ | 0.99 | 0.96 | 0.99 | 0.99 | 0.993 | 0.99 | 0.993 | 0.993 |
| $\gamma_3$ | 0.993 | 0.97 | 0.993 | 0.993 | 0.995 | 0.993 | 0.995 | 0.995 |
| **Cost range** | **150** | **80** | **40** | **140** | **200** | **80** | **100** | **100** |
| Thresholds | | | | | | | | |
| Small | 30 | 30 | 10 | 30 | 40 | 30 | 30 | 30 |
| Middle | 70 | 50 | 20 | 70 | 100 | 50 | 50 | 50 |
| High | 110 | 70 | 40 | 110 | 160 | 70 | 70 | 70 |

## B.3. Additional Experiments

To further validate the superiority of our proposed method, we incorporate seven additional complex tasks on SafetyGymnasium (Ji et al., 2023). Specifically, we select the four "Point" and three "Velocity" tasks as additional benchmarks different from Bullet-Safety-Gym. We compare C2IQL with two well-performed baselines in Table 1: FISOR and CDT. As shown in Table 6, C2IQL achieves the best performance while satisfying constraints on all tasks. Among most tasks, C2IQL achieves nearly 10% improvement compared with other safe performance.

To further demonstrate the contributions of the cost reconstruction model and constraint-conditioned method to C2IQL, we include training curves for task *DroneCircle* as an additional ablation study in Section 4.2. As shown in Figure 7, only C2IQL achieves the highest rewards while satisfying the constraint. C2IQL w/o CC and C2IQL w/o CR fail to satisfy the constraint for small or large constraints, and are more conservative in other cases compared with C2IQL. CIQL is overly conservative in all cases even though it is much safer.

*Table 6.* Normalized evaluation results on more complex tasks. The normalized cost threshold is set to 1. Values in (.) represent the standard deviation. Each value represents the average performance over 10 evaluation episodes with 5 seeds and 3 thresholds. **Black** indicates safe results; gray indicates unsafe results; and blue indicates safe and best-performing results. ↑ (↓) incidates that higher (lower) values are better.

| Algorithm | Metric | Task | | | | | | | |
| | | Point | | | | Velocity | | | |
| | | Circle1 | Goal1 | Push1 | Button1 | HalfCheetah | Hopper | Ant | Avg |
| FISOR | reward ↑ | $0.65_{(0.04)}$ | $\mathbf{0.61}_{(0.01)}$ | $\mathbf{0.24}_{(0.02)}$ | $\mathbf{0.06}_{(0.01)}$ | $\mathbf{0.86}_{(0.01)}$ | $\mathbf{0.15}_{(0.02)}$ | $\mathbf{0.83}_{(0.00)}$ | $\mathbf{0.49}$ |
| | cost ↓ | $1.23_{(0.44)}$ | $\mathbf{0.48}_{(0.03)}$ | $\mathbf{0.11}_{(0.08)}$ | $\mathbf{0.02}_{(0.02)}$ | $\mathbf{0.00}_{(0.00)}$ | $\mathbf{0.03}_{(0.02)}$ | $\mathbf{0.00}_{(0.00)}$ | $\mathbf{0.27}$ |
| CDT | reward ↑ | $\mathbf{0.56}_{(0.04)}$ | $0.69_{(0.01)}$ | $\mathbf{0.25}_{(0.02)}$ | $0.52_{(0.02)}$ | $\mathbf{0.98}_{(0.02)}$ | $\mathbf{0.72}_{(0.02)}$ | $\mathbf{0.99}_{(0.00)}$ | $\mathbf{0.67}$ |
| | cost ↓ | $\mathbf{0.63}_{(0.23)}$ | $1.03_{(0.26)}$ | $\mathbf{0.45}_{(0.12)}$ | $1.63_{(0.87)}$ | $\mathbf{0.06}_{(0.04)}$ | $\mathbf{0.42}_{(0.24)}$ | $\mathbf{0.46}_{(0.12)}$ | $\mathbf{0.67}$ |
| C2IQL | reward ↑ | $\mathbf{0.67}_{(0.09)}$ | $\mathbf{0.75}_{(0.01)}$ | $\mathbf{0.34}_{(0.02)}$ | $\mathbf{0.34}_{(0.02)}$ | $\mathbf{0.99}_{(0.02)}$ | $\mathbf{0.79}_{(0.04)}$ | $\mathbf{1.00}_{(0.01)}$ | $\mathbf{0.56}$ |
| | cost ↓ | $\mathbf{0.77}_{(0.11)}$ | $\mathbf{0.92}_{(0.05)}$ | $\mathbf{0.46}_{(0.05)}$ | $\mathbf{0.91}_{(0.33)}$ | $\mathbf{0.51}_{(0.11)}$ | $\mathbf{0.34}_{(0.08)}$ | $\mathbf{0.58}_{(0.18)}$ | $\mathbf{0.64}$ |

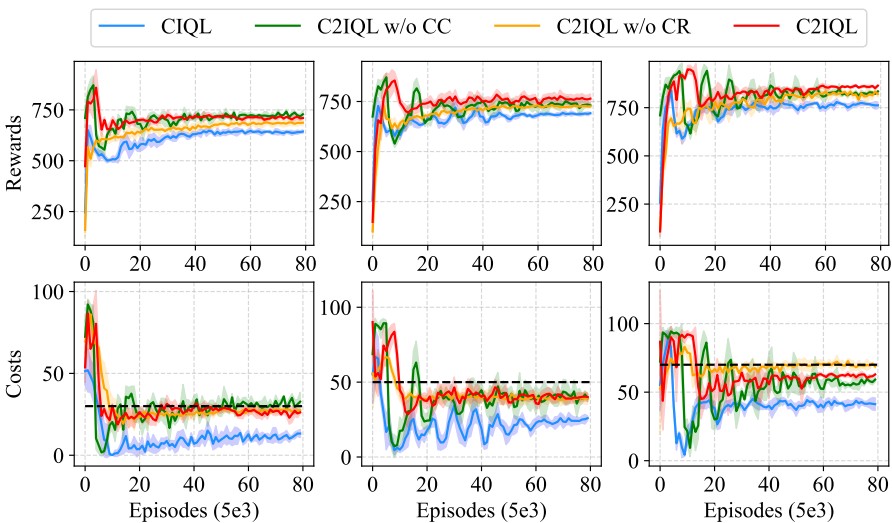

*Figure 7.* Ablation study on *DroneCircle* with varying constraint thresholds. The three plots from left to right represent different threshold levels: small = 30, middle = 50, and high = 70. The dashed lines in each plot indicate the corresponding threshold value for that case.

## B.4. Infrastructure and Runtime Analysis

Experiments are carried out on NVIDIA GeForce RTX 3080 GPUs. We have done the runtime analysis about C2IQL and corresponding baselines, as shown in Table 7. We follow the same iteration times, which is 400,000, for most algorithms. As for VOCE and WSAC, we follow the same setting with the corresponding works and make sure of their convergence. Overall, the training cost of C2IQL is reasonable when compared to other methods. While it takes slightly more time than methods like FISOR and VOCE, it is still significantly faster than CDT, which has the highest training time. Notably, C2IQL achieves a remarkable balance between computational efficiency and performance. The additional training time is justified by the significant performance gains provided by C2IQL, making it a practical and effective solution.

*Table 7.* Runtime analysis of C2IQL and corresponding baselines on task *AntCircle*.

| Algorithm | BCQ-Lag | Bear-Lag | COptiDICE | CPQ | FISOR | VOCE | CDT | WSAC | C2IQL |
|---|---|---|---|---|---|---|---|---|---|
| Training Iterations | 400,000 | 400,000 | 400,000 | 400,000 | 400,000 | 40,000 | 400,000 | 30,000 | 400,000 |
| Training Time | 4h23min | 5h08min | 2h57min | 3h46min | 2h05min | 2h33min | 11h32min | 3h42min | 5h41min |

