# OpenReview forum: "C2IQL: Constraint-Conditioned Implicit Q-learning for Safe Offline Reinforcement Learning"
_ICML.cc/2025/Conference — ICML 2025 poster_

### Official Review · Reviewer_qPih · 2025-02-25

**Overall Recommendation:** 3

**Summary:**

This paper considers offline CMDP problem where one needs to maximize the expected cumulative reward subject to the constraint that the expected total cost is below a certain threshold. The paper then proposed constrained conditioned implicit Q learning for safe offline RL. The main novelty is redefining the reward function in a way that only when the policy is safe it is getting reward.  Empirical results have shown that their result is better compared to the state-of-the-art CDT results in most cases.

**Claims And Evidence:**

The paper seeks to show the value of the constrained condition implicit Q learning for finding safe policies with good reward. The main difficulty is how to train the new model. In particular, they use a novel reward and a novel cost reconstruction for different parameters of the discount function. The claim is that they achieve a better policy compared to the state-of-the-art approaches has been backed up by empirical results.

**Essential References Not Discussed:**

The main drawback of this paper is that essential references have not been discussed. In particular, the paper did not compare with the following paper [A1] which also proposed a novel approach for the safe offline RL and has shown better results compared to CDT.

Further, [A1] considered the following objective $V_r^{\pi}+\lambda (V_{c}^{\pi}-b)_+$,

hence, only when the policy is feasible it adds to the objective, otherwise, it is not which is in the similar flavor of the approach proposed in the paper. It is true that this paper uses the Implicit Q learning, however, [A1] shows provable safety and sub-optimality gap. **Overall, the contribution part is not clear.**


[A1]. Wei, Honghao, Xiyue Peng, Arnob Ghosh, and Xin Liu. "Adversarially Trained Weighted Actor-Critic for Safe Offline Reinforcement Learning." In The Thirty-eighth Annual Conference on Neural Information Processing Systems.

**Experimental Designs Or Analyses:**

Experimental design is solid. However, the improvement over the CDT approach is very minimal. See the questions for detailed explanation.

**Methods And Evaluation Criteria:**

They used detailed experimentation on well-known benchmark problems to validate their claims. The evaluation criteria are the rewards attained and whether the policy is safe or not. I do not find any weakness in the evaluation criteria.

**Other Comments Or Suggestions:**

**Post-Rebuttal**

I am happy with the responses provided by the authors and am more confident about the contributions. Thus, I have raised my score. I would encourage the authors to add those discussions (especially, the need for reconstruction of cost, and the augmentation of the state) in the final version. It would be super important.

**Other Strengths And Weaknesses:**

Weakness:

The improvement compared to the CDT seems to be marginal. Hence, it is not clear the benefits of the proposed approach.

The cost reconstruction part is not clear.

The paper did not provide any theoretical guarantee on the safety or the sub-optimality gap.

**Questions For Authors:**

1. How did the paper reconstruct the cost by finding the value function corresponding to different discount factors?

2. The overall idea is to solve equation (4). However, how is it related to (3), should the reward be $0$ if the policy is not feasible? In particular, the indicator function should be multiplied with the reward as well.

3. The paper has 4 different loss functions which might be tedious for training.

**Relation To Broader Scientific Literature:**

The paper seeks to contribute to the space of the safe offline RL algorithm. However, the paper has not compared with one of the most prominent literature. Hence, the contribution part is not clear.

**Theoretical Claims:**

The two theoretical results are proved. They are pretty straightforward and there is no issue.

---

> ### Author Rebuttal · Authors · 2025-03-31
>
> ### Q1: The main novelty is redefining the reward function that only gets rewards when the policy is safe. The contribution part is not clear.
> In SORL, the most important concern is the OOD problem, while existing methods can only mitigate it via policy constraining but cannot avoid it. The first novelty of this paper is proposing C2IQL to completely avoid the OOD problem in SORL.
>
> The second novelty is that we may be the first one to propose and analyze the problem of discounted cost value formulation in detail. The contribution is illustrated in line 87, page 2 while redefining the reward function is the novelty of CPQ (2022).
>
> For better clarity, we will add new content to line 88 page 2 and line 100 page 2:
>
> "We first propose Constrained IQL (CIQL) to address the OOD problem, which can only be mitigated for existing RL methods in constrained settings."
>
> "As far as we know, we may be the first one to propose and analyze the problem of discounted cost value formulation in detail."
> ### Q2: References [A1] is not discussed/compared.
> Thank you for bringing this paper to our attention. We agree that [A1] is relevant to SORL, and we will include it in the related work section and as a baseline in the experiments. Specifically, we will add the following to line 205, page 4:
>
> "... problem. WSAC (Wei et al., 2024) focuses on the policy inferior problem and proposes a refined adversarial objective function. OASIS ..."
>
> Additionally, we compared WSAC with C2IQL across all environments. The results are summarized below where bold results are unsafe:
>
> |Algo||AR|BR|CR|DR|AC|BC|CC|DC|Avg|
> |-|-|-|-|-|-|-|-|-|-|-|
> |WSAC|R|0.25|0.804|0.86|0.659|0.40|0.69|0.61|0.024|0.537|
> ||C|0.179|**1.98**|0.40|**2.518**|0.98|0.78|0.51|0.45|0.97|
> |C2IQL|R|0.74|0.59|0.95|0.71|0.66|0.72|0.74|0.78|0.74|
> ||C|0.94|0.95|0.08|0.73|0.76|0.85|0.93|0.85|0.76|
>
> C2IQL achieves better performance while satisfying constraints on all tasks.
> ### Q3: The improvement compared to the CDT seems to be marginal.
> The performance gain only appears marginal because **the unsafe results in smaller constraints are averaged by safe results in larger constraints** since Table 1 is averaged out 3 constraint thresholds following the style of most existing work. Thus we add Figure 2 as a supplement of Table 1 to illustrate that:
> 1. CDT **cannot achieve safe policy for small constraints (like L<30)** under some cases. This is very fatal disadvantage since satisfying the constraint is the foundation of safety.
> 2. CDT **cannot achieve reward maximization for large constraints (like L<70)** compared to C2IQL.
> 3. C2IQL achieves best and safe performance for all three constraints. This indicates C2IQL provides substantial benefits in handling a wider range of constraints and reward maximization accordingly.
> ### Q4: The cost reconstruction part is not clear. How did the paper reconstruct the cost by finding the value function corresponding to different discount factors?
> We would like to provide the following clarifications:
> The definition of the cost value function is a linear function where $\gamma$  is the constant and  $c(.,.)$ are variables.
>
> $V_c^\pi(s_t)=\sum_{j=0}^T\gamma^jc(s_{t+j},a_{t+j}\sim\pi)$
>
> The goal is to reconstruct the non-discounted cost value:
>
> $\hat C^{\pi}(s_t)=\sum_{j=0}^Tc(s_{t+j},a_{t+j}\sim\pi)$
>
> To approximate the non-discounted cost value, we construct a set of cost value functions with different discount factors:
>
> $V_c^{\pi,i}(s_t)=\sum_{j=0}^T\gamma^j_ic(s_{t+j},a_{t+j}\sim\pi),i=1,...,m$
>
> Using these multiple discounted value functions, we train a supervised learning model to estimate the non-discounted cost value, where the inputs are estimated $V_c^{\pi,i}(s_t)$ and the output is $\hat C^{\pi}(s_t)$.
> ### Q5: Should provide theoretical guarantee.
> Please refer to **Q2 of reviewer MiHo** for theoretical analysis due to character limit. The mathematical proof are out of the scope of this paper. Besides, C2IQL has good empirical results including additional experiments on SafetyGymnasium from **Q1 reviewer ojm5**.
> ### Q6: Question about equation (3) and (4).
> Yes, equation (3) lets the infeasible reward Q-value to be 0 and equation (4) indicates only feasible policy is updated, which are the key ideas of CPQ (2022). However, this is not what C2IQL should be focused on. While your suggestion to multiply the indicator function with the reward is reasonable and could improve CPQ, it is outside the focus of our work, as we adopt CPQ’s framework without modification. Investigating this idea further would be interesting for future work.
> ### Q7: About 4 different loss functions.
> Compared with existing safe RL algorithms, we only add one loss to train the cost reconstruction model (CRM). In safe RL, value loss, cost value loss, and policy loss are essential. For the CRM, it is trained separately with the dataset and some randomly generated data. It is used in C2IQL after the loss is minimized and stable. Thus the training stage is not influenced.

---

> > ### Comment · Reviewer_qPih · 2025-04-02
> >
> > I would like to thank the authors for their responses. The new results seem promising.
> >
> > I have a few more comments:
> >
> > 1. It seems that the approach is based on CPQ. The main claim of this paper is that CPQ cannot address the OOD actions. The paper then considers Implicit Q learning proposed for unconstrained cases, in the constrained setting. Hence, the algorithmic contributions seem to be limited. Can the authors highlight that?
> >
> > 2. The standard CMDP approach admits a Markov optimal policy. However, the CPQ approach seems to be like an augmented MDP where the Q function for the constraint is augmented with the state space in order to find a policy. Hence, conceptually, it is more complicated. Thus, it is not clear why one needs such complicated state augmentation to solve the CMDP problem.

---

> > > ### Author Response · Authors · 2025-04-03
> > >
> > > We sincerely thank reviewer `qPih` for the engagement. We are encouraged by the review, which described our work as novel, with solid experiments and presenting promising new results. For the additional comments, we would like to provide the following clarifications:
> > >
> > > ### Q1:
> > >
> > > Thank you for your question. First, we would like to highlight that addressing the OOD issue is not our only contribution. Additionally, we study the problem of discounted cost formulation in detail and then develop a novel approach for reconstructing non-discounted costs.
> > >
> > > Regarding the OOD avoidance, we address the following three key technical challenges:
> > >
> > > - **First, how to update the constrained reward value/Q-value function following IQL style?** To address this problem, CIQL formulates a constraint-penalized reward Q-value function following CPQ and utilize a value function with expectile regression to approximate the maximized Q-value function in Bellman backup procedure.
> > > - **Second, how to update the cost value function under the same implicit policy since it is hidden in the reward value function?** To address this problem, we rederive CIQL and obtained the formulation of the implicit policy in theorem 1 following IDQL, and then derive the formulation of the cost value function with this implicit policy.
> > > - **Third, How to extract the policy?** We extract the policy in an expectile way following Equation 18."
> > >
> > > Besides, we think the clarifications in Q2 can further answer your questions about Q1 very carefully. We will illustrate why the idea of CPQ is needed and what is the main novel for building CIQL rather than simply converting IQL to the constrained settings.
> > >
> > > ### Q2:
> > >
> > > This is a good question and we would like to provide the following discussion: **The main claim of this paper is that most SORL methods cannot avoid the OOD problem, including but not limited to CPQ. This motivates us to propose a method that can completely avoid the OOD problem** so we resort to IQL, which is an algorithm that can avoid OOD problems in unconstrained settings.
> > >
> > > **However, IQL cannot be simply extended to constrained settings due to the following challenges:**
> > > IQL proposes to approximate $\max_aQ(s,a)$ by a value function with expectile regression without any explicit policy. The key idea to avoid OOD problem is that the policy is implicitly hidden in the value function during training.  After the value function is trained well, IQL utilize a policy extraction method to obtain the final explicit policy.
> > >
> > > As we know, Safe RL usually contains two value function: reward value function and cost value function following the same policy. Thus the main gap is that IQL utilizes a implicit policy hidden in the reward value function but the cost value function should following the same implicit policy in constrained settings. We address this gap in our paper by answering:
> > >
> > > **How to make sure both value functions follows the same policy without facing the OOD problem?**
> > >
> > > Existing standard CMDP methods, such as primal-dual optimization (PDO), cannot address this problem. In PDO (like BCQ-Lag), an explicit policy is needed to make sure both primal and dual objectives follows the same policy. However, an explicit policy will result in the OOD problem in offline RL during Bellman backup. Thus explicitly extracting the policy of IQL out and then utilizing this policy to update cost value function is not reasonable because the OOD problem is introduced for cost value function.
> > >
> > > **This motivates us to find a method that satisfies: (1) both value functions follows the same policy and (2) both value functions follows the implicit policy.**
> > >
> > > One advantage of CPQ (like augmented MDP) is that the update of the reward value function and policy are not tightly tied together. This provides us a possibility to satisfy (1) and (2) together. Thus we first formulate the constraint-penalized reward value function (**Equation 11, 12 and 13**) for CIQL following the idea of CPQ. Then the problem becomes how to update cost value function under the same implicit policy hidden behind the reward value function. To address this problem, we rederive the representation of CIQL to a generalized AC structure (**Equation 14**) via first order optimality condition (**Appendix A.1**) to obtain the representation of the implicit policy. **Following the implicit policy, we construct how to update the coat value function under the same implicit policy (Equation 15, 17) without extracting it out explicitly.**
> > >
> > > We hope these responses and additional results provided in our rebuttal address your concerns and encourage you to consider a more favorable evaluation of our paper. Thank you again for the time you investigated in evaluating our paper.

---

### Official Review · Reviewer_QM2i · 2025-03-14

**Overall Recommendation:** 3

**Summary:**

The paper introduces Constraint-Conditioned Implicit Q-Learning (C2IQL), a novel approach for Safe Offline Reinforcement Learning (SORL) that improves constraint satisfaction while maximizing rewards. The key innovations include a Cost Reconstruction Model (CRM), which estimates non-discounted cumulative costs to improve safety, and constraint-conditioned learning, which allows policies to dynamically adapt to different safety thresholds. The authors provide theoretical results supporting their approach and conduct experiments on Bullet-Safety-Gym tasks, demonstrating that C2IQL outperforms existing safe RL baselines in terms of both constraint satisfaction and reward maximization.

## update after rebuttal
I appreciate the authors’ detailed response and the additional experiments. The new SafetyGymnasium results and the clarifications on the theoretical derivations (particularly how Theorem 1 supports cost value updates in Theorem 2) help address my main concerns. The runtime and baseline setup explanations are also helpful. In short, I find the core contributions meaningful for the SORL community. I maintain my score.

**Claims And Evidence:**

1. The claim that C2IQL improves constraint satisfaction is supported by experiments where it achieves better safety-performance tradeoffs than baselines.
2. The claim that CRM improves cost estimation is justified through its design, but there is no formal analysis of how errors in cost reconstruction affect policy learning.
3. The paper suggests that C2IQL mitigates the Out-of-Distribution (OOD) problem, but there is no theoretical guarantee on how well it generalizes to unseen cost distributions beyond the dataset.

**Essential References Not Discussed:**

N/A

**Experimental Designs Or Analyses:**

Advantages:
1. The experiments demonstrate that C2IQL outperforms baselines in both reward maximization and constraint satisfaction, showing its effectiveness in safe offline RL settings.
2. The evaluation covers a range of environments from Bullet-Safety-Gym, providing insights into how C2IQL handles safety constraints in practical scenarios.

Limitations:
1. While the results show improvements, the study does not analyze how well the cost reconstruction model generalizes to unseen cost distributions.
2. The results in Table 1 are comprehensive but a bit difficult to interpret, as they only provide final values without learning curves. A visualization of policy evolution would improve clarity.

**Methods And Evaluation Criteria:**

1. The use of Bullet-Safety-Gym tasks is reasonable for evaluating safety-constrained RL.
2. The choice of baselines (BCQ-Lag, BEAR-Lag, COptiDICE, CPQ, FISOR, CDT, and VOCE) is appropriate for comparing constraint satisfaction and reward maximization

**Other Comments Or Suggestions:**

1. Algorithm 2 is a bit difficult to interpret, as the equations are dense. The authors could consider placing key equations in an appendix or breaking them down incrementally within the main text to improve clarity. A more structured explanation before introducing the full algorithm would help readers understand its components more intuitively.
2. I wonder if the heavy use of abbreviations in the abstract and introduction is necessay, as it affects readability.

**Other Strengths And Weaknesses:**

1. The impact of cost reconstruction errors on policy learning is not analyzed, making it unclear how estimation inaccuracies influence performance.
2. There is no evaluation of sample efficiency, leaving the question of how much data C2IQL requires compared to baselines unanswered.

**Questions For Authors:**

See the above sections.

**Relation To Broader Scientific Literature:**

1. The paper builds on prior work in offline RL (IQL, COptiDICE) and safe RL (CDT, VOCE).
2. The CRM approach is novel compared to prior pessimistic offline RL approaches.

**Theoretical Claims:**

1. Theoretical results provide constraint satisfaction guarantees and introduce a cost reconstruction model (CRM) to improve estimation of non-discounted cumulative costs. The approach enhances policy learning by refining constraint enforcement without requiring explicit policy extraction.
2. However, the method relies on the assumption that cost reconstruction provides sufficiently accurate estimates, and the paper does not formally analyze how errors in this model may propagate through policy learning. A deeper analysis of how inaccuracies in cost estimation impact safety guarantees could strengthen the theoretical foundation.

---

> ### Author Rebuttal · Authors · 2025-03-31
>
> ### Q1: An experiment of how errors/inaccuracies in cost estimation impact safety guarantees is needed.
> Thank you for your suggestion. To address your concern, we have conducted additional experiments to analyze the impact of cost reconstruction error on policy performance and safety guarantees.
> To simulate increasing cost reconstruction error, we introduced noise sampled from a normal distribution, $N(0,Noise)$, to the input of the cost reconstruction model. When the added noise is small, the performance (reward) slightly improves as the noise level increases. This occurs because small amounts of noise act similarly to data augmentation, which enhances the generalization capability of the cost reconstruction model. This is analogous to adding noise to input data in image processing to improve robustness. As noise increases from 1.0 to 4.0, the performance becomes progressively more conservative. Despite this, the reward and cost metrics degrade gracefully, indicating the robustness of our method.
> | Noise | 0.0 | 0.1 | 0.2 | 0.3 | 0.5 | 1.0 | 2.0 | 4.0 |
> | - | - | - | - | - | - | - | - | - |
> | Error | 0.05 | 1.05 | 1.33 | 1.50 | 1.83 | 2.1 | 2.4 | 2.8 |
> | R $\uparrow$ | 434.01 | 435.07 | 442.08 | 435.67 | 435.60 | 421.82 | 408.77 | 401.89 |
> | C $\downarrow$ | 46.60 | 47.20 | 48.30 | 47.30 | 47.89 | 42.10 | 34.90 | 36.20 |
>
> ### Q2: The paper suggests that C2IQL mitigates the OOD problem, there is no theoretical guarantee on how well it generalizes to unseen cost distributions beyond the dataset.
> Thank you for your question.
>
> 1, we would like to clarify that C2IQL **avoids the OOD problem rather than merely mitigating it**. In SORL, the OOD problem is defined as "Since the policy may produce OOD actions, the Q-value may be wrongly estimated". Previous methods address it by constraining the target policy to be close to behavior policy, thus only mitigating the problem. In contrast, C2IQL, following IQL, completely avoids the OOD problem by training the value function using expectile regression, which inherently focuses on in-distribution data during training.
>
> 2, **C2IQL does not require generalization to unseen cost distribution** because: 1) C2IQL avoids the OOD problem entirely by ensuring that the training process operates strictly within the dataset distribution. When training, C2IQL avoids relying on any distributions outside the dataset. Even during testing phase, C2IQL naturally tends to select actions that align with the dataset distribution, as it is trained to do so. 2) As for the cost reconstruction model (CRM), they will not encounter unseen cost distribution because CRM is only used during training. In testing phase, CRM is no longer utilized—only the policy is used to generate actions based on the current state and constraint conditions.
> ### Q3: The results in Table 1 are a bit difficult to interpret without learning curves.
> Thank you for your suggestions. We will add some more learning curves to the appendix.
> Table 1 follows a common style in most SORL papers (e.g., CDT, FISOR, OASIS), where results are presented as final values after training. Additionally, we use color coding to improve interpretability.
> The primary reason for not including learning curves is due to the nature of offline RL. In SORL, the agent is trained on a pre-collected and fixed dataset without interacting with the environment during training. As a result, monitoring performance during intermediate training steps is not typically necessary, since the agent's performance is only evaluated after the training process is complete.
> That said, we have included some learning curves in Figure 3 to provide a visualization of policy evolution.
> ### Q4: Evaluation of sample efficiency is needed.
> Thank you for your suggestions. In Offline RL, the agent is trained on a fixed and pre-collected dataset without any interactions with environments. This indicates that all methods (C2IQL and baselines) require the same amount of data without any new data added into the dataset. The dataset in DSRL usually contains thousands of trajectories.
> ### Q5: Algorithm 2 is a bit difficult to interpret, which could consider to improve clarity.
> Thank you for your suggestion. Algorithm 2 is indeed a collection of equations previously introduced in Sections 3.1–3.4, but now extended to the constraint-conditioned version. We realize that we did not explicitly indicate their connections to the earlier sections, which may have caused confusion.
>
> To improve clarity, we will provide additional explanations in Algorithm 2 to the corresponding sections and equations:
>
> Line 6: Cost Reconstruction -> Obtain discounted cost values and reconstruct the non-discounted value with $R^c$ in Section 3.3
>
> Line 9: Constraint Penalization -> Obtain the constraint-penalized Q-value function with Equation (11)
>
> Line 11: Update value function -> Update value function with Equation (12)
>
> Line 14: Update cost value function -> Update cost value function with Equation (17)

---

> > ### Comment · Reviewer_QM2i · 2025-04-03
> >
> > Thank you for the detailed and thoughtful rebuttal. I appreciate the additional experiments and clarifications provided, particularly the analysis of cost reconstruction errors and the explanation on how C2IQL handles the OOD issue. Your responses addressed many of my concerns, and I am looking forward to the planned improvements.

---

> > > ### Author Response · Authors · 2025-04-03
> > >
> > > We sincerely thank the reviewer `QM2i` for the engagement. We are delighted that the reviewer recognized the novelty and effectiveness of our proposed method in their initial review. In response to reviewer `QM2i`'s suggestions, we plan to incorporate the following updates to the next version:
> > >
> > > 1. Adding experimental analysis on the impact of cost reconstruction error on policy performance and safety guarantees.
> > > 2. Including additional learning curves to improve clarity.
> > > 3. Refining the manuscript by incorporating our rebuttal changes and addressing the following issues:
> > >    - Improve Algorithm 2 for clarity:
> > >      - Line 6: Cost Reconstruction -> Obtain discounted cost values and reconstruct the non-discounted value with $R^c$ in Section 3.3
> > >      - Line 9: Constraint Penalization -> Obtain the constraint-penalized Q-value function with Equation (11)
> > >      - Line 11: Update value function -> Update value function with Equation (12)
> > >      - Line 14: Update cost value function -> Update cost value function with Equation (17)
> > >    - Less Abbreviations, for example:
> > >      - Delete "SORL" in line 16 and 21, page 1 in abstract because the first sentence already determine the scope as SORL
> > >      - Delete "ACPO" in line 70, page 2
> > >      - Delete "CMDP" in line 72, page 2

---

### Official Review · Reviewer_ojm5 · 2025-03-14

**Overall Recommendation:** 3

**Summary:**

This work focuses on the offline safe RL, where the existing baseline methods often suffer from the OOD issues (as in the general offline RL). To do so, this paper proposes the C2IQL method that employs the cost reconstruction model to derive non-discounted cumulative
costs from discounted values and incorporates a flexible, constraint-conditioned mechanism to accommodate dynamic safety constraints. This work also provides some empirical evidence on the Bullet-Safety-Gym environments.

## update after rebuttal

My concerns has been addressed by the authors and I have updated my score.

**Claims And Evidence:**

The experimental results are less convincing. Please see the weakness and questions below.

**Essential References Not Discussed:**

N/A

**Experimental Designs Or Analyses:**

Some of the experimental results are less convincing and shows limited improvements than the baselines. Please see the weakness below.

**Methods And Evaluation Criteria:**

The benchmark environments are commonly considered in OSRL. However, the authors use Bullet-Safety-Gym only. Other more complicated environments like SafetyGymnasium, MetaDrive should be considered as well.

**Other Comments Or Suggestions:**

1. I suggest that the authors should focus on C2IQL, maybe no need to say that much about CIQL, especially maybe don't treat it as a new method but part of the C2IQL?

**Other Strengths And Weaknesses:**

1. The idea is interesting.

2. I think the main weakness is that the proposed method is only evaluated on Bullet-Safety-Gym, which is relatively simple. DSRL has other more complicated environments like SafetyGymnasium, MetaDrive.

3. There are no discussions/comments for the Theorems.

4. How do the authors obtain the results of baselines, e.g., in Table 1? a) The proposed method has very limited improvement to be honest. b) The proposed method is not beating the baselines that are mentioned in other OSRL work, e.g., see Table 5 of the below work

Gong, Z., Kumar, A., & Varakantham, P. (2024). Offline Safe Reinforcement Learning Using Trajectory Classification. arXiv preprint arXiv:2412.15429.

**Questions For Authors:**

N/A

**Relation To Broader Scientific Literature:**

N/A

**Theoretical Claims:**

Theorems look reasonable, but I didn't carefully check the proofs.

---

> ### Author Rebuttal · Authors · 2025-03-31
>
> ### Q1: More complex benchmark is needed.
> Thank you for your suggestion. To address the reviewer’s suggestion, we have also incorporated additional experiments on the SafetyGymnasium benchmark to diversify the test tasks. Specifically, we have selected the 4 Point and 3 Velocity tasks as additional benchmarks different from Bullet-Safety-Gym. Given limited time, we have included results of C2IQL and two strong baselines (FISOR and CDT). The bold results are **unsafe** cases.
> |Env||C2IQL|CDT|FISOR|
> |-|-|-|-|-|
> |PointCircle1|R|0.666 (0.09)|0.562 (0.04)|0.651 (0.04)|
> ||C|0.772 (0.11)|0.628 (0.23)|**1.230 (0.44)**|
> |PointGoal1|R|0.745 (0.01)|0.689 (0.01)|0.612 (0.01)|
> ||C|0.924 (0.05)|**1.033 (0.26)**|0.474 (0.03)|
> |PointPush1|R|0.343 (0.02)|0.252 (0.02)|0.244 (0.02)|
> ||C|0.464 (0.05)|0.447 (0.12)|0.114 (0.08)|
> |PointButton1|R|0.343 (0.09)|0.519 (0.02)|0.063 (0.01)|
> ||C|0.913 (0.33)|**1.631 (0.87)**|0.018 (0.02)|
> |HalfCheetahVelocity|R|0.985 (0.02)|0.975 (0.02)|0.855 (0.01)|
> ||C|0.512 (0.11)|0.061 (0.04)|0 (0)|
> |HopperVelocity|R|0.791 (0.04)|0.715 (0.02)|0.153 (0.02)|
> ||C|0.344 (0.08)|0.422 (0.24)|0.029 (0.02)|
> |AntVelocity|R|0.996 (0.01)|0.986 (0)|0.832 (0)|
> ||C|0.575 (0.18)|0.46 (0.12)|0 (0)|
>
> The additional evaluation on SafetyGymnasium demonstrates that our method, C2IQL, achieves nearly a 10% improvement in performance for most tasks while satisfying constraints, further validating its superior performance.
> ### Q2: No discussions for Theorems.
> Thank you for your suggestion. We will an explanation of Theorem 1 in line 255 to illustrate how it can be utilized in Theorem 2:
> "Theorem 1 provides us with a relationship between the constraint-penalized reward function and corresponding formulation of the implicit policy. When the implicit policy formulation follows Equation (14), the update of the value function under implicit policy $\pi_{imp}(a|s)$ is equivalent to the update formulation of the value function in Equation (13) under behavior policy (the policy for collecting the dataset). Thus we can utilize the formulation of the implicitly policy obtained in Theorem 1 to derive how to update the cost value function based on its definition in Theorem 2."
> ### Q3: How do the authors obtain the results of baselines?
> For BCQ-Lag, BEAR-Lag, COptiDICE, CPQ, and CDT, we use the code provided by the OSRL library. For FISOR and VOCE, we use the official code released by the respective papers. To ensure a fair comparison, we run each method across 10 evaluation episodes, using 5 random seeds and 3 different constraint thresholds.
> ### Q4: The proposed method has very limited improvement to be honest.
> The performance gain only appears marginal to CDT because the unsafe results in smaller constraints is averaged by safe results in larger constraints since Table 1 is averaged out 3 constraint thresholds following the style of most existing work. Thus we add Figure 2 as supplement of Table 1 to illustrate that:
> 1. CDT **cannot achieve safe policy for small constraints (like L<30)** under some tasks. This is very fatal disadvantage since satisfying the constraint is the foundation of safety.
> 2. CDT **cannot achieve reward maximization for large constraints** (like L<70) compared to C2IQL.
> 3. C2IQL achieves **best and safe performance for all three constraints**. This indicates C2IQL provides substantial benefits in handling a wider range of constraints and reward maximization accordingly.
> ### Q5: The proposed method is not beating baselines in other OSRL work [1].
> Thank you for bringing up this concurrent work. We would like to address the reviewer’s concern in the following aspects:
>
> **Baseline Algorithm Selection Criteria:** We carefully select strong-performing methods (e.g, CDT (2023), VOCE (2023), FISOR (2024), WSAC (2024) from reviewer qPih) from recent publications in SORL field to ensure a comprehensive comparison. Extensive experiments on Bullet-Safety-Gym and SafetyGymnasium benchmarks have demonstrated the superior performance of our proposed C2IQL. Regarding the concurrent work [1], we excluded it from our comparison because it was only posted on arXiv in 12/2024 and has not undergone the peer review process.
>
> **Compare with [1]:**
> 1. **Performance**: By directly comparing the results from Table 5 in [1] and Table 1 in our paper, we observe that C2IQL provides stronger performance compared to [1]:
> - C2IQL achieves better results on AR, AC, BR, BC, CC, DR, and DC, demonstrating its strong performance across a variety of tasks.
> - C2IQL exhibits slightly worse performance on CarRun compared to [1].
> 2. **Method**: [1] addresses issues with poorly behaved policies under global cost constraints. In contrast, C2IQL focuses on the OOD problem and is the first method designed to avoid the OOD problem in SORL. Besides, we also address the discounted cost formulation problem, which further improves constraint satisfaction and policy effectiveness.
>
> [1] Offline Safe Reinforcement Learning Using Trajectory Classification.

---

> > ### Comment · Reviewer_ojm5 · 2025-04-03
> >
> > Thank you for your response and for your additional experiments. I have a few comments below.
> >
> > 1. How many random seeds do the authors consider in the additional experiments?
> >
> > 2. As indicated by Table 1, FISOR is not a strong baseline. BCQ-Lag has very good avg reward even though it slightly violates the constraint (I assume it may work for the new SafetyGymnasium tasks). Even COptiDICE and CPQ outperform FISOR as shown in Table 1.
> >
> > 2. I didn't mean that the authors had to compare exactly with paper "Offline Safe Reinforcement Learning Using Trajectory Classification". This is just an example indicating that the baseline methods in other papers could be better than the baseline results shown in this paper. Plus that the improvements of this work is limited. Then it is hard to see if this work really achieves an outperforming results than the current SOTA methods.

---

> > > ### Author Response · Authors · 2025-04-03
> > >
> > > We sincerely thank reviewer `ojm5` for the engagement and appreciation of our interesting idea. We provide the following clarifications to address the remaining questions:
> > > ### Q1:
> > > We use 5 random seeds for each case, which is consistent with Table 1 in our paper. As the results show, the performance gain of C2IQL clearly exceeds the variance margin, demonstrating significant improvement over existing baselines. Details in: https://anonymous.4open.science/r/rebuttal-B9B9/README.md
> > > ### Q2:
> > > Thank you for your comments. We want to clarify that:
> > > 1. **FISOR is indeed a strong baseline** as it achieves safety across all tasks, which is a fundamental requirement for SORL algorithms.
> > > 2. BCQ-L's higher average performance comes at the cost of safety violations. Specifically, BCQ-Lag violates constraints in 4/8 scenarios, indicating insufficient performance in safety-critical environments. We would like to emphasize that **comprehensive evaluation of SORL algorithms requires examining both safety satisfaction and reward performance.**
> > > 3. BCQ-L is added to the link and performs unsafely in SafetyGymnasium.
> > > ### Q3:
> > > We agree on the importance of fair and comprehensive comparisons with current SOTA methods in SORL to demonstrate the superior performance of C2IQL. And we believe we have adequately done so by:
> > > 1. **Including relevant SOTA methods:** We conducted a comprehensive survey of recent publications (including ICML, ICLR, NIPS) in 2023-2024 and incorporated **enough methods demonstrating promising results [2-6]** as baselines in Table-R1. Extensive experiments on Bullet-Safety-Gym and SafetyGymnasium show the superior performance of C2IQL, establishing it as the new SOTA.
> > > 2. **Ensuring faithful baseline reproduction**: We utilized official implementations or reliable third-party code to evaluate baseline performance. We also cross-compared our reproduced results with those originally reported (please see Table-R2) and found our reproduction achieves comparable or better performance for baseline methods, ensuring fair comparison.
> > > 3. **Incorporating the additional baseline** suggested by the reviewer `ojm5` and `qPih`. Results show C2IQL outperforms [1] and WSAC, further validating our method's effectiveness.
> > >
> > > **Additional baseline reproduction details:**
> > >
> > > Table-R1. Published paper in SORL field and corresponding baselines compared in the paper.
> > > |Paper/Baseline|BCQ-L|BEAR-L|COptiDICE|CPQ|CDT|FISOR|VOCE|WSAC|
> > > |-|-|-|-|-|-|-|-|-|
> > > |This paper|Y|Y|Y|Y|Y|Y|Y|Y|
> > > |CDT(2023)[2]|Y|Y|Y|Y|Y|N|N|N|
> > > |VOCE(2023)[3]|Y|N|Y|N|N|Y|Y|N|
> > > |FISOR(2024)[4]|Y|N|Y|Y|Y|Y|N|N|
> > > |WSAC(2024)[5]|Y|Y|Y|Y|N|N|N|Y|
> > > |OASIS(2024)[7]|Y|Y|Y|Y|Y|Y|N|N|
> > > |[1] (2024)|Y|Y|Y|Y|Y|N|N|N|
> > >
> > > Table-R2 below cross-references our reproduced results with those in recent publications:
> > >
> > > Table-R2. Reported results comparison of baselines in recently published papers. For method doesn't compared on Bullet-safety-gym, we just use "N" to show. For those tested on Bullet-safety-gym, we list the performance shown in the original paper.
> > >
> > > |Baseline|BCQ-L||BEAR-L||COptiDICE||CPQ||CDT||FISOR||VOCE||WSAC||
> > > |-|-|-|-|-|-|-|-|-|-|-|-|-|-|-|-|-|
> > > |**Paper**|R|C|R|C|R|C|R|C|R|C|R|C|R|C|R|C|
> > > |C2IQL|0.69|1.05|0.49|1.49|0.53|0.80|0.41|0.83|0.71|0.91|0.31|0.06|0.42|1.80|0.54|0.97|
> > > |CDT|0.79|2.67|0.59|1.2|0.50|2.96|0.76|2.84|0.83|0.72|N|N|N|N|N|N|
> > > |VOCE|N|N|N|N|N|N|N|N|N|N|N|N|N|N|N|N|
> > > |FISOR|0.71|10.63|N|N|0.55|7.64|0.32|5.28|0.63|2.09|0.39|0.1|N|N|N|N|
> > > |WSAC|0.51|1.12|0.52|1.43|0.36|1.44|0.36|1.63|N|N|N|N|N|N|N|N|
> > > |OASIS|0.78|3.21|0.65|4.38|0.64|2.30|0.34|9.07|0.63|2.44|0.41|0.48|N|N|N|N|
> > > |[1]|0.74|3.11|0.48|3.8|0.55|2.55|0.33|1.12|0.68|1.04|N|N|N|N|N|N|
> > >
> > > Performance variations across papers can be attributed to:
> > > 1. Different constraint threshold settings (FISOR employs nearly zero constraints, resulting in relatively large normalized costs)
> > > 2. Task selection differences (CDT omits complex AntCircle (S=34, A=8, T=500))
> > > 3. Different problem focuses (OASIS addresses dataset mismatch problems and omits AntRun and AntCircle; it is essentially BCQ-Lag + diffusion-based data augmentation)
> > >
> > > We hope these responses and additional results provided in our rebuttal address your concerns and encourage you to consider a more favorable evaluation of our paper. Thank you again for the time you investigated in evaluating our paper.
> > >
> > > [2] Liu, Z. et al., 2023. Constrained decision transformer for offline safe reinforcement learning. ICML
> > >
> > > [3] Guan, J. et al., 2023. Voce: Variational optimization with conservative estimation for offline safe reinforcement learning. NIPS
> > >
> > > [4] Zheng, Y.et al, 2024. Safe Offline Reinforcement Learning with Feasibility-Guided Diffusion Model. ICLR
> > >
> > > [5] Wei, H. et al., 2024. Adversarially Trained Weighted Actor-Critic for Safe Offline Reinforcement Learning. NIPS
> > >
> > > [6] Xu, H. et al., 2022. Constraints penalized q-learning for safe offline reinforcement learning. AAAI
> > >
> > > [7] Yao, Y.et al., 2024. Oasis: Conditional distribution shaping for offline safe reinforcement learning. NIPS

---

### Official Review · Reviewer_MiHo · 2025-03-17

**Overall Recommendation:** 3

**Summary:**

Offline RL has gained popularity recently as it can be trained with offline batched data without interacting with a simulation environment. Constrained offline RL further extends the idea by adding threshold penalty on the entire trajectory cost. While offline RL and constrained on-policy RL are relatively well studied problems, constrained offline RL still remains a challenging problem. The paper proposed to solve this problem by taking ideas from IQL and IDQL (to address OOD problem) and introducing a cost reconstruction model to address the problem with discounted cost in offline setting. Experimental results on Bullet-Safety-Gym toy simulation environment demonstrate that the proposed C2IQL can outperform other baseline methods in terms of improving the reward values while maintaining the cost budget.

**Claims And Evidence:**

Constrained offline RL is a challenging problem. The paper claims to solve this problem with cost reconstruction and IQL method. The major claims in improving the reward and cost satisfaction have been proven theoretical and experimentally.

**Essential References Not Discussed:**

NA

**Experimental Designs Or Analyses:**

Experiments are designed carefully with standard practice on safety-gym environments. C2IQL is also compared with several SOTA techniques. Ablation studies were conducted to demonstrate the value for different steps proposed in C2IQL. Overall, the results seem complete, but adding some results on real-world problems with noisy offline transition data would have been more appreciated.

**Methods And Evaluation Criteria:**

The proposed evaluation criteria in safety-gym environment are a standard practice in offline RL problems. The evaluation criteria also make sense to me. My only concern is that the threshold penalty is chosen randomly in experiments. Adding experiments on real-world problem settings or a simulated version of some real problem would have been appreciated.

**Other Comments Or Suggestions:**

Overall, the paper is written well and easy to follow. However, it is hard to follow the theoretical results and algorithms, so adding a brief high-level summary will be helpful to readers.

**Other Strengths And Weaknesses:**

Constrained offline RL is a challenging problem and the paper addressed the main challenges such as OOD problem and discounted cost reconstruction is a valuable contribution. Experimental results also seem to cover the key areas of evaluation. Here are some concerns though:
1. The motivation behind constraint offline RL is somewhat missing. Adding a few real-world examples in the introduction will be helpful.
2. Experiments are only conducted in simulation environment with randomly generated cost trajectories and threshold. Therefore, it is hard to predict how such techniques will perform in real-world scenarios with noisy transition sample trajectories.
3. Convergence proof and computation complexity or runtime analysis is largely missing.

**Questions For Authors:**

1. Please provide a few real-world examples of constrained offline RL.
2. Have you done any runtime analysis of C2IQL? Can you provide the results for different methods considered in experiments.
3. Have you finetuned parameters of different baseline algorithms or just took some random configuration?

**Relation To Broader Scientific Literature:**

Constrained offline RL is not a well explored problem yet, and the real-world applicability seems to be in niche areas. Therefore, it would be of interest to a limited scientific community.

**Theoretical Claims:**

Theorem 1 and 2 (extended from IQL) gives reasonable insights on how to derive the algorithm, but quality guarantees and convergence proofs are missing.

---

> ### Author Rebuttal · Authors · 2025-03-31
>
> ### Q1: My only concern is that the threshold penalty is chosen randomly. Adding experiments on real-world problem settings is appreciated.
> Thank you for your suggestion. The threshold penalty is choose based on small (<50%), middle (50%), and large (>50%) constraint of the maximum cost, which is comprehensive to cover real-world constraint settings.
> Besides, we sincerely agree that addressing real-world problems is important. However, in most existing SORL benchmarks [1, 2], there are nearly no real-world simulators since SORL is a newly rising field in recent years. Thus we will consider it in our future work.
> To address your concern, we have expand experiments on other environments in [2]. Please refer to Q1 of **reviewer ojm5** for detailed results due to character limit.
>
> [1] Gronauer, S. (2022). Bullet-safety-gym: A framework for constrained reinforcement learning.
>
> [2] Ji, J., Zhang, B., Zhou, J., Pan, X., Huang, W., Sun, R., ... & Yang, Y. (2023). Safety gymnasium: A unified safe reinforcement learning benchmark. *Advances in Neural Information Processing Systems*, *36*, 18964-18993.
> ### Q2: Theorem 1 and 2 miss quality guarantees and convergence proofs.
> We appreciate the reviewer recognizing the insights provided by theorem 1 and 2, which highlight our contribution avoid the OOD problem and address the discounted formulation problem in constrained settings. Regarding the quality guarantees and convergence proofs, we think our method share the same results as CPQ theoretically. The sketch of the proof is given as follows: First, we have that as the expectile parameter $\kappa$ increase to 1,  $V^\pi_r(s)$ can approach $\max_aQ^\pi_{r|c}(s,a)$ because the expectile regression is a convex function. Second, convergence proofs of our algorithm can follow similar steps as those in CPQ because CPQ utilize $\max_aQ^\pi_{r|c}(s,a)$ while C2IQL utilize $V^\pi_r(s)$.
> ### Q3: Adding a few real-world examples to motivate constraint offline RL in introduction.
> Thank you for your suggestion. We add a few real-world examples on the right part of line 24, page 1, second paragraph, to further strengthen our motivation:
> "… in safety-critical scenarios. **For example, unsafe operations could harm patients in healthcare, unsafe driving style may lead to accidents, and unsafe decisions may incur additional costs in financial investments.** In these situations, …"
> ### Q4: Adding a brief high-level summary will be helpful to follow the theoretical results and algorithms.
> Thank you for your suggestion. We will add a high-level summary at the beginning of section 3.1:
> "To derive a concrete CIQL algorithm, we need to answer three questions: **First, how to update the constrained reward value/Q-value function following IQL style?** To address this problem, CIQL formulates a constraint-penalized reward Q-value function following CPQ and utilize a value function with expectile regression to approximate the maximized Q-value function in Bellman backup procedure. **Second, how to update the cost value function under the same implicit policy since it is hidden in the reward value function?**  To address this problem, we rederive CIQL and obtained the formulation of the implicit policy in theorem 1 following IDQL, and then derive the formulation of the cost value function with this implicit policy. **Third, How to extract the policy?** We extract the policy in an expectile way following Equation 18."
> ### Q5: Runtime analysis is needed.
> We record the training time of our proposed C2IQL and baselines in AntCircle scenario in the following table:
> | Algorithm     | BCQ-Lag | Bear-Lag | COptiDICE | CPQ     | FISOR   | VOCE    | CDT      | C2IQL   |
> | ------------- | ------- | -------- | --------- | ------- | ------- | ------- | -------- | ------- |
> | Training Time | 4h23min | 5h08min  | 2h57min   | 3h46min | 2h05min | 2h33min | 11h32min | 5h41min |
>
> Overall, the training cost of C2IQL is reasonable when compared to other methods. While it takes slightly more time than methods like FISOR and VOCE, it is still significantly faster than CDT, which has the highest training time. Notably, C2IQL achieves a remarkable balance between computational efficiency and performance. The additional training time is justified by the significant performance gains provided by C2IQL, making it a practical and effective solution
> ### Q6: About baseline configuration?
> All baseline algorithms follow the default parameters, which are already finetuned in OSRL [1] projects for best performance. As for FISOR and VOCE, we finetuned the parameter to make sure the performance keeps consistent with the original paper. However, under the same environment but different constraints, we keep the same parameters for all methods including our proposed C2IQL.
>
> [1] Liu, Z., Guo, Z., Lin, H., Yao, Y., Zhu, J., Cen, Z., ... & Zhao, D. (2023). Datasets and benchmarks for offline safe reinforcement learning. *arXiv preprint arXiv:2306.09303*.

---

### Decision · Program_Chairs · 2025-05-01

**Decision:**

Accept (poster)

**Comment:**

This paper addresses the offline learning problem in constrained Markov Decision Processes (MDPs), where the objective is to maximize the expected cumulative reward while ensuring that the expected total cost remains below a specified threshold. The novel contribution is a Cost Reconstruction Model (CRM), which estimates non-discounted cumulative costs, allowing policies to adapt dynamically to varying safety thresholds.

Overall, the reviewers agree that this work is technically sound and has the potential to contribute to the existing reinforcement learning literature. However, they noted that the simulations were conducted in a simple synthetic environment, raising concerns about the applicability of the results to more complex settings. Additionally, there was insufficient discussion regarding the proposed theorems. The authors have partially addressed these concerns by including additional experiments on more complex environments and providing a more detailed comparison with baseline methods during their discussion. It is recommended that the authors incorporate these additional details into the camera-ready manuscript.